# GameSR: Real-Time Super-Resolution for Interactive Gaming

## Abstract

High-resolution gaming demands significant computational resources, with challenges further amplified by bandwidth and latency constraints in cloud gaming. Existing upscalers, such as NVIDIA DLSS and AMD FSR, reduce rendering costs but require engine integration, making them unavailable for most titles, especially those released before the introduction of upscalers. We present **GameSR**, a lightweight, engine-independent super-resolution model that operates directly on encoded game frames. The architecture of GameSR combines reparameterized convolutional blocks, PixelUnshuffle, and a lightweight ConvLSTM to deliver real-time upscaling with high perceptual quality. Extensive objective and subjective evaluations on popular games, such as *Counter-Strike 2*, *Overwatch 2*, *FIFA* and *Team Fortress 2*, show that GameSR reduces cloud gaming bandwidth usage by 30–60% while meeting target perceptual qualities, achieves real-time performance of up to 240 FPS, substantially outperforms existing super-resolution models in the literature, and reaches near-parity with DLSS and FSR *without* accessing rendering engine data structures or modifying game source code, making GameSR a practical solution for upscaling both modern and legacy games with no additional development effort.

## 1 Introduction

Gaming is the world's largest entertainment industry, surpassing film and music with revenues of over $200 billion in 2024 and projected to reach nearly $290 billion by 2030 (CAGR 8.7%) (Statista, 2025; Newzoo, 2024). High-resolution, high-frame-rate gaming is highly immersive but computationally demanding. As resolution and frame-rate requirements increase (e.g., 2K and 4K at 60–120 fps), the processing cost rises sharply. For example, high-end GPUs such as the RTX 3080 Ti have power ratings of $\sim$350 W, and measurements of games like *Horizon Zero Dawn* confirm draws of $\sim$346 W under full load (Igor's Lab, 2021; Tom's Hardware, 2021). System-level tests further show that demanding modern titles can push full gaming PCs to 325–380 W at ultra 4K settings (Mezha, 2024). Combined with analyses estimating that gaming rigs can consume $\sim$1,400 kWh/year under heavy use (Mills & Mills, 2015), these figures highlight the substantial hardware and operational costs required to sustain premium gaming performance.

An emerging alternative to this hardware-intensive model is cloud gaming, where games are rendered on remote servers and streamed to lightweight clients. While this shifts the computational burden away from players, it introduces substantial bandwidth and latency challenges. Unlike video streaming services such as Netflix, which stream 1080p content at around 5 Mbps (3 GB/hr) (Netflix, 2020; 2022), platforms like Nvidia Gforce Now demand at least 28 Mbps for 1080p (12.6 GB/hr) (NVIDIA, 2025), due to fast motion, complex animations, and latency-sensitive compression profiles (e.g., small GOPs and no B-frames). Moreover, gaming is highly interactive, requiring round-trip response within milliseconds to preserve player performance and Quality of Experience (QoE). Prior studies show first-person shooter games tolerate up to 80 ms end-to-end latency (Amiri et al., 2020), while every additional 100 ms can reduce third-person game performance by 25% (Claypool & Finkel, 2014). Latency arises from client input, server rendering/encoding, and network delay; the latter alone can consume up to 80% of the total budget (Choy et al., 2012).

A common way to reduce rendering costs is to lower spatial resolution and then upscale; however, naive upscaling degrades visual quality. Hardware vendors have therefore introduced content-aware

solutions such as NVIDIA DLSS (NVIDIA, 2019), AMD FSR (AMD, 2025), and Intel XeSS (Intel, 2024). While effective, these upscalers require game engine integration and access to depth maps, motion vectors, and other internal data structures, with additional vendor restrictions (e.g., DLSS on NVIDIA hardware only). Research models like RenderSR (Dong et al., 2022), ExtraSS (Wu et al., 2023b), Mob-FGSR (Yang et al., 2024), and Neural Supersampling (Xiao et al., 2020b) follow the same tightly coupled approach. As a result, support remains limited to a small subset of modern titles, leaving legacy engines and forward-rendered pipelines unable to adopt these upscalers.

In contrast, a large body of work on super-resolution for *general* images and videos (e.g., (Lim et al., 2017; Lai et al., 2017; Hui et al., 2019; Luo et al., 2020; Liang et al., 2021)) can operate directly on rendered frames without requiring game-engine integration. While these models achieve good upscaling quality, they are typically too slow for interactive use, with inference times far exceeding real-time budgets, as confirmed by our experiments in §4. As such, these models remain unsuitable as a general-purpose upscaling solution for gaming.

The goal of this paper is to introduce a video game upscaler that reduces computing cost while preserving high visual fidelity, and that operates independently of the game engine without requiring source code. Achieving this is challenging: strict latency constraints leave little tolerance for extra processing, most industrial upscalers rely on engine-level data (e.g., motion vectors, depth), and any solution must be lightweight enough to coexist with rendering, encoding, and networking in real time. Even minor overheads risk stutter or added input-to-display delay, as modern pipelines already push frame budgets to the limit, often disabling effects like motion blur or ambient occlusion at higher frame rates. Thus, an effective upscaler must be engine-agnostic, efficient, and carefully integrated to deliver perceptual gains without breaking interactivity. We present evaluations in §4, with additional results and implementation details provided in Appendix A due to space constraints.

The main contributions of this paper are as follows.

- We propose GameSR (§3.2), a lightweight neural super-resolution model that operates directly on rendered frames **without requiring access to game source code or game engine data structures**, making it readily deployable in cloud gaming for recent and legacy games.
- We demonstrate that GameSR achieves **near-parity with industrial upscalers** on no-reference perceptual metrics, despite using no motion vectors or depth buffers (§4.2).
- GameSR matches SOTA quality while running **30–60× faster** than CNN baselines and nearly **500× faster** than SwinIR, with up to an **order-of-magnitude smaller** size and memory (§4.2).
- We demonstrate that streaming at lower resolutions and upscaling with GameSR yields **30–60% bandwidth savings** while meeting various perceptual quality targets. (§4.3).

## 2 BACKGROUND AND RELATED WORK

**Stand-alone Gaming and Upscalers.** Most games run locally on PCs or consoles, where detailed textures, fast motion, and complex effects like ray tracing demand powerful GPUs. To reduce load, super-resolution (SR) methods render at lower resolutions or frame rates and then upscale the frames, a process that is far cheaper than full-resolution rendering.

Industry solutions include DLSS (NVIDIA, 2022), FSR (AMD, 2022), and XeSS (Intel, 2022). DLSS uses autoencoder and transformer-based models, FSR applies adaptive interpolation with post-processing passes, and XeSS employs deep learning. While effective, all require integration into the game source code to access engine data such as motion vectors, depth, and color, which complicates deployment and limits applicability.

Academic work has also advanced real-time upsampling. Neural Supersampling (Xiao et al., 2020a) leverages depth and motion vectors but suffers from ghosting in dynamic scenes; Li et al. (Li et al., 2024) separate lighting and material components for better temporal stability; and ExtraSS (Wu et al., 2023a) combines spatial supersampling with frame extrapolation via G-buffer–guided warping. Like industrial solutions, these approaches also rely heavily on the game engine data structures.

**Limitations of Engine-Integrated Upscalers.** The reliance on engine data structures limits the applicability of existing upscalers to a narrow set of modern titles. Legacy games, many of which

still have active communities, are particularly excluded. For instance, Team Fortress 2, released in 2007 on Valve's original Source engine, has not been ported to the modern Source 2 pipeline and therefore cannot expose the motion vectors, depth buffers, or temporal anti-aliasing required by DLSS 2/3 and FSR 2/3 (AMD GPUOpen, 2025; NVIDIA, 2025). Similar restrictions apply to other forward-rendered games, such as Counter-Strike 2, where the rendering pipeline lacks temporal data that upscalers depend on. As a result, despite the large catalog of PC games, DLSS, FSR, and XeSS are only supported in a limited subset of titles for which developers have explicitly integrated them (NVIDIA, 2025; AMD, 2025). In fact, while Steam alone hosts over 86,000 games (SQ Magazine, 2025), only about 650 titles support DLSS (NVIDIA, 2025) and roughly 350–400 support FSR (AMD, 2025), i.e., well under 1% of the catalog. Furthermore, Steam itself does not represent the entire ecosystem; other major platforms such as the Epic Games Store, PlayStation Store, and Xbox Marketplace host thousands of additional titles, making the relative coverage of current upscalers even smaller in the broader gaming landscape.

**Suitability of Existing Image/Video Upscalers for Gaming.** Prior work has proposed numerous image and video SR models, including EDSR (Lim et al., 2017), LapSRN (Lai et al., 2017), IMDN (Hui et al., 2019), LatticeNet (Luo et al., 2020), and SwinIR (Liang et al., 2021). More recently, research has focused on real-time SR, exploring architectural refinements (Andrey Ignatov et al., 2021), model compression, and novel training methods to balance quality with reduced computation, parameters, and memory (Ignatov et al., 2022; Li et al., 2022; Conde et al., 2023).

Lightweight SR models improve efficiency through various strategies: IMDN uses information distillation, RFDN replaces it with feature distillation connections, and FMEN emphasizes inference optimization with tuned convolutions and re-parameterization. LapSRN employs a Laplacian pyramid for coarse-to-fine upsampling, while LatticeNet integrates residual and attention mechanisms to halve parameters without quality loss. However, these designs target general efficiency rather than the millisecond-level latency demands of cloud gaming, which remain unmet (details in §A.1).

To quantify this gap, we evaluate existing SR models on gaming content in §4.2. Our results show that even IMDN (Hui et al., 2019), the most efficient among them, takes over 120 ms to upscale a single frame by $2\times$ on a high-end GPU, far exceeding real-time limits. By contrast, GameSR takes 4.1 ms on the same hardware.

Additionally, Recurrent video SR methods such as RLSP Fuoli et al. (2019), MRVSR Chiche et al. (2022), and SSL pruned BasicVSR Wang et al. (2023) use heavier recurrent backbones with fixed $4\times$ scaling on small inputs (for example $180\times320$ to 720p) and report runtimes of tens of milliseconds per frame on high end GPUs. In contrast, GameSR targets $2\times$ and $3\times$ upscaling of full HD game streams (for example 540p or 720p to 1080p) in about 4 to 5 milliseconds per frame with only 138K parameters, which is a more suitable operating point for real time local and cloud gaming.

**Additional Challenges of Cloud Gaming.** In cloud gaming, rendering is done on the cloud, and the resulting frames are streamed to clients. Since clients receive only compressed video streams, industrial upscalers, as well as rendering-coupled research models (Dong et al., 2022; Wu et al., 2023b; Yang et al., 2024; Xiao et al., 2020b; Meyer et al., 2022; Zheng et al., 2025; Zhong et al., 2023; Yang et al., 2023; Zhang et al., 2024), cannot be applied: they depend on motion vectors, depth, and other engine-level data unavailable at the client side. Moreover, even if executed in the cloud, such methods would not reduce streaming bitrate, since frames must still be transmitted at display resolution.

Finally, while no SR approaches have been specifically designed for cloud gaming, video-on-demand (VOD) streaming has explored SR integration (Yeo et al., 2018; Baek et al., 2021; Yeo et al., 2020). These frameworks pre-train lightweight "micro" models for each video segment and transmit them alongside the stream. However, this is infeasible in *interactive* cloud gaming systems, where frames are generated in real time based on player inputs.

**Feasibility of Running Upscalers on Client Devices.** Most client devices used for gaming sessions possess underutilized compute resources capable of running upscalers. For example, smartphones such as iPhone 16 Pro (Apple A18 Pro, 35 TOPS) and MediaTek Dimensity 9400 (50 TOPS) include powerful NPUs, while consoles like the PS5 and Xbox Series X offer over 10 TFLOPS of GPU compute (Apple Inc., 2024; MediaTek Inc., 2024; Sony Interactive Entertainment, 2024; Microsoft, 2024). A naive port of heavy VSR models to these devices is still impractical due to power, thermal, and latency constraints. GameStreamSR Bhuyan et al. (2024) addresses this by upscaling only a

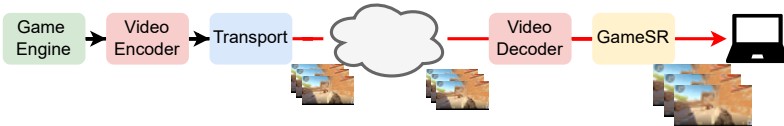

Figure 1: GameSR in cloud gaming: low-resolution streams are rendered at server-side and upscaled at client-side in real time.

depth-defined region of interest on mobile and using bilinear interpolation for the rest of the frame, but this engine-dependent, RoI-only strategy limits quality outside the focus area and assumes access to render buffers. In contrast, our goal is to exploit this latent client compute with a lightweight, full-frame, engine-agnostic upscaler that can run within a few milliseconds per frame on both mobile and desktop hardware, and is compatible with post-decoder cloud-gaming pipelines where only compressed RGB video is available.

# 3 PROPOSED SOLUTION

## 3.1 OVERVIEW AND OPERATION

We design GameSR as an engine-independent, lightweight super-resolution (SR) model that can be utilized in both traditional (stand-alone) and cloud gaming systems. In traditional gaming, GameSR can be applied as a post-processing step after frames are rendered by the game engine, enhancing the frames before they are displayed.

In contrast, in cloud gaming, the game engine renders frames on the server, which are then compressed by the video encoder and transmitted over the network. This is illustrated in Figure 1. On the client side, the decoder reconstructs compressed frames, which are normally displayed directly to the player. To upscale frames in real time, GameSR is inserted between the decoder and the display, transparently improving the quality of the frames as they arrive.

In addition to improving perceived quality for players, GameSR offers three advantages for cloud gaming: (i) it reduces server rendering and encoding load by allowing operation at lower resolutions, (ii) it lowers transmission bitrate since fewer pixels are streamed, and (iii) it requires no integration with the game engine or decoder internals, making it readily deployable for recent and legacy games.

These placements make GameSR both engine-agnostic and codec-agnostic: any title or streaming service that outputs an RGB video stream can benefit from the same model, without modifying the game engine, exposing render buffers, or depending on a specific vendor's upscaling stack.

The key challenge of designing GameSR is meeting the strict deadline in highly interactive gaming environments. We illustrate the high-level design of GameSR in Figure 2. As our evaluation in §4 demonstrates, GameSR improves perceived quality while meeting real-time latency requirements of gaming. We present the details of various components of GameSR in the following.

## 3.2 GAMESR DETAILS

We design GameSR as a lightweight SR model for latency-sensitive gaming content, with neural layers and components specifically designed for efficiency and effectiveness. GameSR, and SR models in general, reconstructs high-resolution (HR) frames from low-resolution (LR) inputs by optimizing a parameterized function $F$ as follows:

$$\theta^* = \arg\min_{\theta} \sum L\left(F(y^{\text{LR}}; \theta), y^{\text{HR}}\right). \tag{1}$$

Fundamentally, the function $F$ performs three main tasks in super-resolution problems: Feature Representation, Feature Learning, and Mapping LR frames to HR ones. In our design, we extend this formulation by introducing a fourth stage—*Temporal Learning*—which leverages information from adjacent frames before the final mapping stage. We summarize each of these tasks in the following. More details can be found in §A.2.

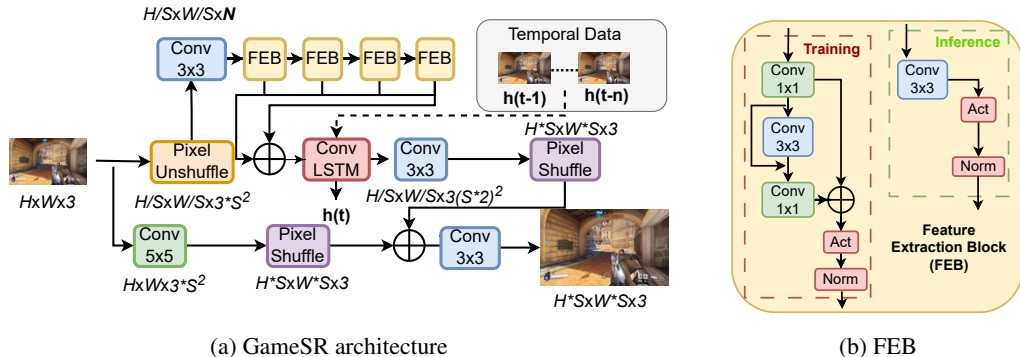

(a) GameSR architecture          (b) FEB

Figure 2: Overview of GameSR: (a) architecture where frames are downsampled with PixelUnshuffle, processed by Feature Extraction Blocks, and passed through a lightweight ConvLSTM before upsampling via PixelShuffle with residual connection; (b) internal structure of FEB.

While the formulation in Eq.1 is general and GameSR could in principle be applied to other video domains, our design is motivated by the unique characteristics of gaming content. As opposed to traditional multimedia, gaming video is synthetic and exhibits recurring objects, structured environments, and repetitive motion patterns (Zadtootaghaj et al., 2018). These properties enable per-game, data-centric training and make it possible to realize an extremely lightweight SR model that still achieves high perceptual quality.

**Feature Representation for Upscaling.** The feature representation stage employs a single $3 \times 3$ convolution and PixelUnshuffle (space-to-depth) (Shi et al., 2016), reducing spatial dimensions by a factor of $s$ and expanding channels by $s^2$. Unlike conventional SR methods (Hui et al., 2019; Liu et al., 2020; Du et al., 2022), this down-and-up scheme significantly reduces computational cost while capturing richer channel-wise feature relationships. A detailed inference time analysis is presented in §4.2. The formulation is:

$$F_1(y^{LR})_{f_1 \times \frac{H}{s} \times \frac{W}{s}} = \max\left(0, W_1 * \text{PixelUnshuffle}(y^{LR}) + B_1\right), \qquad (2)$$

where $W_1$ and $B_1$ are convolution weights and biases. PixelUnshuffle with scale factor $s$ rearranges the input from $c \times H \times W$ into $c \cdot s^2 \times \frac{H}{s} \times \frac{W}{s}$, increasing the channel dimension by $s^2$ while reducing spatial dimensions. These features are then passed into Feature Learning blocks.

**Feature Learning.** The feature learning stage captures non-linear mappings between LR and HR features using our Feature Extraction Block (FEB), which is shown in Figure 2b. During training, each FEB applies a $1 \times 1 \rightarrow 3 \times 3 \rightarrow 1 \times 1$ convolution sequence, expanding and then compressing feature dimensions. At inference, we merge these into a single convolution via reparameterization (Deng et al., 2023), greatly reducing computational load without accuracy loss.

Each FEB incorporates GeLU activations and LayerNorm (Ba et al., 2016) for stable and efficient training. Residual connections preserve spatial detail and facilitate gradient flow. After sequential FEBs, we employ multi-level feature aggregation through additive fusion, defined as:

$$RB_{final} = \sum_{i=0}^{N} RB(i), \qquad (3)$$

where each FEB output is combined additively, enhancing gradient propagation, feature reuse, and memory efficiency. The aggregated features are then fed into a lightweight ConvLSTM to capture temporal information

**Temporal Learning.** Video super-resolution (VSR) leverages temporal information across frames to enhance quality, making it especially relevant for gaming sequences in cloud gaming. Unlike single-image SR, VSR exploits motion continuity through either explicit (e.g., optical flow (Dosovitskiy et al., 2015)) or implicit alignment (e.g., 3D/deformable convolutions (Ying et al., 2020; Shi et al., 2022)). However, most VSR models are too computationally heavy for real-time deployment.

To balance temporal modeling and efficiency, we adopt a lightweight variant of ConvLSTM (Shi et al., 2015) after feature extraction. ConvLSTM replaces matrix multiplications in standard LSTMs with convolutions, preserving spatial resolution while capturing long-range dependencies. Our design uses a single-layer structure with decoupled gates (input, forget, output, and cell), each implemented with independent 2D convolutions. This modular design enables better parallelization on modern GPUs while minimizing sequential overhead.

During inference, frames are processed sequentially using hidden states from prior frames, enabling effective motion-aware upsampling. The ConvLSTM operates over spatial features with dimensions $(C, H/s, W/s)$ and uses standard gate updates:

$$i_t = \sigma(W_i * [x_t, h_{t-1}] + b_i), \quad f_t = \sigma(W_f * [x_t, h_{t-1}] + b_f), \tag{4}$$

$$o_t = \sigma(W_o * [x_t, h_{t-1}] + b_o), \quad \tilde{c}_t = \tanh(W_g * [x_t, h_{t-1}] + b_g), \tag{5}$$

$$c_t = f_t \odot c_{t-1} + i_t \odot \tilde{c}_t, \quad h_t = o_t \odot \tanh(c_t). \tag{6}$$

Here, $*$ denotes 2D convolution and $[\cdot, \cdot]$ is channel-wise concatenation. As shown in §4.2, this temporal module significantly improves perceptual quality under motion. Finally, the temporally enhanced features are upsampled back to display resolution.

**Mapping from Low to High Resolutions.** Our upsampling stage utilizes Pixel-shuffling for spatial resolution enhancement, avoiding checkerboard artifacts common in deconvolution methods (Odena et al., 2016; Long et al., 2015). This approach reshapes feature channels into spatial dimensions efficiently. We incorporate a residual connection by combining upsampled ConvLSTM output with the original input, preserving fine details and textures. Formally, the operation is expressed as:

$$\hat{y}^{SR}_{c \times H \times W} = \text{Conv}(\text{PixelShuffle}(\max(0, W_{up} * (RB)_{f_1 \times H \times W} + B_{up}))). \tag{7}$$

## 4 EVALUATION

### 4.1 SETUP AND PERFORMANCE METRICS

**Games.** We evaluate on three distinct games: Counter-Strike 2 (CS2), Overwatch 2 (OW2), and Team Fortress 2 (TF2). CS2 and OW2 represent modern, high-demand titles, while TF2 serves as a legacy case. Using VirtualDub (Lee, 2024), we captured uncompressed 1080p gameplay at 30/60 FPS across diverse maps, motions, and lighting. Five players of varying skill levels recorded five sessions per game (25 sessions total), yielding 40k frames for CS2, 54k for OW2, and 30k for TF2. We used 10 sessions per game for training and 15 unseen sessions for testing.

**Performance Metrics.** We evaluate quality using commonly used metrics: PSNR, SSIM, VMAF (Netflix, 2018), and LPIPS (Ghazanfari et al., 2023). PSNR/SSIM are pixel-based, while VMAF/LPIPS better capture perceptual quality. In gaming, reference frames are often unavailable due to engine non-determinism, floating-point variability, multithreaded scheduling, and event-driven randomness, which prevent frame-level consistency (Chance et al., 2022). Thus, we also employ two no-reference models: NDNetGaming (Utke et al., 2022), tailored to gaming with MOS-like scores, and VSFA (Li et al., 2019), a ResNet-50+GRU model. These are primarily used for DLSS/FSR comparisons. In addition, we measure bandwidth, and GPU usage.

**Training.** We trained GameSR in PyTorch 2.0.1 on an NVIDIA RTX A4000 with an Intel Xeon Gold 5220 CPU and 32 GB RAM. Training used AdamW (Loshchilov & Hutter, 2019) ($\beta_1$=0.9, $\beta_2$=0.999), learning rate $10^{-3}$ halved every $2 \times 10^5$ iterations, minibatch size 16, and Charbonnier loss. Data was split 80/20 for training/validation. For deployment, we compiled the model with Torch-TensorRT using kernel fusion and mixed precision (FP32 inputs, FP16 kernels) to improve throughput and memory efficiency. Further details are in §A.3.

### 4.2 PERFORMANCE ANALYSIS OF GAMESR

**GameSR Performance.** To evaluate GameSR's performance, we utilized it to upscale diverse gameplay sessions across different maps, users, and character configurations, ensuring a wide range of visual variability. We present sample results in Figure 3 for upscaling sessions from the CS2 and OW2 games by a factor of 2X. As shown in the figure, GameSR consistently achieves high-quality

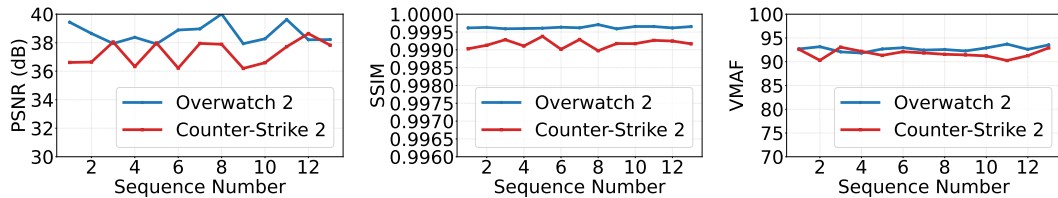

Figure 3: Performance of GameSR on upscaling game sessions from CS2 and OW2 by factor of 2X.

Table 1: Comparing GameSR against DLSS and FSR, which require engine-level data. In contrast, GameSR upscales encoded streams. Results shown for $2\times$ scaling on CS2, OW2, and TF2.

| Model | Counter-Strike 2 | | Overwatch 2 | | Team Fortress 2 | |
|---|---|---|---|---|---|---|
| | NDNetGaming ↑ | VSFA ↑ | NDNetGaming ↑ | VSFA ↑ | NDNetGaming ↑ | VSFA ↑ |
| DLSS | - | - | **4.93** | **0.88** | - | - |
| FSR | **5.00** | **0.89** | 4.81 | 0.81 | - | - |
| GameSR | 4.90 | 0.83 | 4.76 | 0.81 | **4.79** | **0.78** |

results: PSNR ranges from 36–40 dB, SSIM exceeds 0.998, and VMAF scores fall within the 90–95 range—indicating excellent quality (Qin et al., 2019; Elecard, 2023).

**GameSR vs. Commercial DLSS and FSR Upscalers.** We compare GameSR against industry-standard upscalers such as FSR and DLSS. FSR and DLSS were applied using in-game settings, whereas we rendered frames natively at 540p and upscaled them directly using GameSR. This approach was designed to provide a realistic reference point; however, it is essential to note that this setup is not entirely fair to GameSR. While DLSS and FSR have access to additional renderer data (e.g., motion vectors, depth buffers), GameSR relies solely on the input frames for upscaling.

We summarize the comparison results in Table 1. Sample frames produced by the considered upscalers are presented in the §A.4 (figure 7) for visual comparisons. We compared GameSR to FSR (1.0/2.2) and DLSS (3.5) using the same gameplay sequences, maps, and camera paths to ensure a fair comparison. In CS2, we tested GameSR against FSR 1.0 in "Performance" mode ($2\times$ upsampling), matching GameSR's scaling factor.

GameSR scored 4.9 (NDNetGaming) and 0.83 (VSFA), closely trailing FSR's 5.0 and 0.89. For OW2, which supports both FSR 2.2 and DLSS 3.5, we also used $2\times$ upscaling factor. GameSR achieved scores of 4.76 and 0.81, nearly matching FSR (4.81, 0.81) and DLSS (4.93, 0.88).

GameSR achieves near-parity with FSR and DLSS in perceptual quality, with differences of only 0.1 (NDNetGaming) and 0.06 (VSFA) in CS2, and within 0.05 (FSR) and 0.17 (DLSS) for ND-NetGaming in OW2. The engine-independent nature makes it more deployable across platforms. For instance, CS2 employs forward rendering and currently does not support temporal elements required by DLSS 2+ or FSR 2+, meaning those modern upscalers cannot be adopted without changes to the rendering pipeline (Valve, 2024).

Team Fortress 2 serves as a representative legacy title in our evaluation. Like many older games, it has not been updated to modern engines such as Source 2, which restricts compatibility with contemporary upscalers like DLSS and FSR that rely on motion vectors, depth buffers, and temporal anti-aliasing. As a result, TF2 and similar legacy titles cannot natively benefit from these industrial solutions. In contrast, GameSR operates directly on rendered frames without engine-level modifications, delivering high-fidelity upscaling comparable to modern titles and extending the visual longevity of older games while maintaining broad deployability.

Beyond quality, we also measured GPU load and FPS (Figure 4c). Native 1080p runs at ≈123 FPS and ∼100% GPU, whereas 540p+GameSR runs at ≈125 FPS with only ∼82% GPU, since it shades $4\times$ fewer pixels and adds only a lightweight SR pass. DLSS 3.5 (≈140 FPS) and FSR 2.2(≈145 FPS) also render internally at reduced resolution but use the saved budget for higher FPS, so their GPU load stays near 99%; in contrast, 540p+GameSR exposes headroom that, in a cloud setup with server-side rendering and client-side upscaling, can translate into meaningful com-

Table 2: Quantitative comparison between state-of-the-art super-resolution models and GameSR at $2\times$ scaling on four popular games. Evaluated on a workstation with an NVIDIA RTX A4000 GPU.

| Model | Inference (ms) | Counter-Strike 2 | | | Overwatch 2 | | | Team Fortress 2 | | | FIFA24 | | |
|---|---|---|---|---|---|---|---|---|---|---|---|---|---|
| | | PSNR | SSIM | LPIPS | PSNR | SSIM | LPIPS | PSNR | SSIM | LPIPS | PSNR | SSIM | LPIPS |
| Bicubic | - | 32.50 | 0.998 | 0.189 | 35.22 | 0.999 | 0.128 | 34.07 | 0.998 | 0.128 | 31.82 | 0.997 | 0.190 |
| SwinIR | 1971.7 | 35.92 | 0.998 | **0.084** | **40.74** | **0.999** | **0.016** | 40.10 | **0.999** | **0.046** | **34.91** | 0.998 | 0.155 |
| LapSRN | 239.7 | 33.63 | 0.998 | 0.107 | 38.09 | 0.999 | 0.030 | 36.77 | 0.998 | 0.144 | 33.48 | 0.998 | 0.172 |
| EDSR | 160.0 | 35.30 | 0.998 | 0.091 | 40.16 | 0.999 | 0.018 | 39.41 | 0.999 | 0.050 | 34.56 | 0.998 | 0.157 |
| LatticeNet | 154.4 | 35.46 | 0.998 | 0.088 | 40.29 | 0.999 | 0.017 | 39.36 | 0.999 | 0.050 | 34.71 | 0.998 | 0.159 |
| IMDN | 121.2 | 35.38 | 0.998 | 0.089 | 40.33 | 0.999 | 0.018 | 39.36 | 0.999 | 0.050 | 34.71 | 0.998 | 0.154 |
| GameSR | **4.12** | **37.99** | **0.999** | 0.095 | 40.36 | 0.999 | 0.021 | **40.88** | 0.999 | 0.051 | 34.849 | **0.998** | **0.143** |

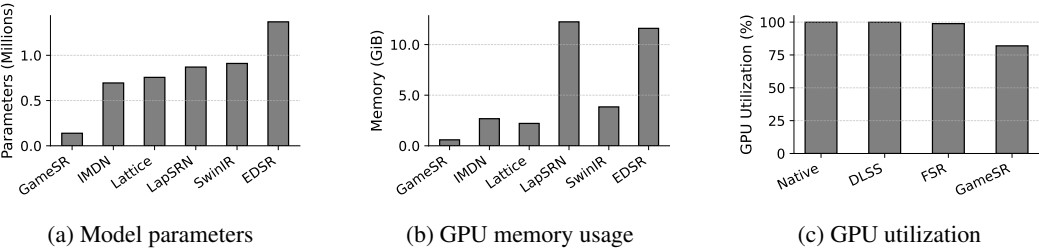

(a) Model parameters        (b) GPU memory usage        (c) GPU utilization

Figure 4: Efficiency of GameSR in comparison to state-of-the-art. Results for $\times 2$ scaling.

pute savings. ***In summary***, GameSR not only saves bandwidth, but it also reduces the computational power needed to render games. This is achieved while providing near-DLSS/FSR quality without accessing the rendering engine's data structures or modifying the game source code.

**GameSR vs. State-of-the-Art Upscalers in the Literature.** To assess the performance and efficiency of our lightweight model, GameSR, we conducted a comparative analysis against several state-of-the-art (SOTA) SR models, including EDSR (Lim et al., 2017), LapSRN (Lai et al., 2017), IMDN (Hui et al., 2019), LatticeNet (Luo et al., 2020), and SwinIR (Liang et al., 2021). In our evaluation, a scaling factor of 2 corresponds to upsampling from 540p→1080p, while a scaling factor of 3 corresponds to 360p→1080p (see § A.5).

Table 2 shows that GameSR matches the quality of state-of-the-art models like SwinIR in PSNR, SSIM, and LPIPS while running orders of magnitude faster. GameSR reaches ∼240+ fps (∼4.1 ms/frame), compared to < 10 fps for EDSR/LatticeNet and < 1 fps for SwinIR, making it practical for real-time cloud gaming. Although SwinIR achieves the highest quality through Transformer-based designs, its heavy cost prevents deployment in latency-sensitive settings. To ensure a fair comparison, we retrained IMDN on our CS2 dataset. As shown in Table 5, GameSR achieves comparable quality with only a 0.18 dB PSNR gap, while being $5\times$ smaller in parameters and $4.5\times$ in memory. We also present model generalization across different games in §A.7

GameSR's efficiency comes from three design choices: ConvLSTM captures temporal dependencies, reparameterization enables wide training but lightweight inference, and PixelUnshuffle reduces spatial cost. Together, these yield real-time performance with high visual fidelity.

Beyond accuracy and runtime, we compared parameter counts and GPU memory across models. As shown in Figure 4(a,b), GameSR uses only 138K parameters and 604 MiB memory, compared to 1.37M/11.9 GiB for EDSR and 910K/3.9 GiB for SwinIR. IMDN and LatticeNet also require 5–6× more memory. At $\times 2$ scale (and similarly at $\times 3$), GameSR achieves order-of-magnitude savings in size and memory over SOTA upscalers.

**Scalability and generality to high-resolution upscaling.** We further evaluate GameSR on 2K (2560×1440) and 4K (3840×2160) CS:GO gameplay sequences. Table 3 reports reconstruction quality and latency for three high-resolution mappings: 720p→2K, 1080p→4K, and 720p→4K. For 1080p→4K, GameSR achieves up to 39.25 dB PSNR, SSIM above 0.999, and VMAF above 93 while keeping inference below 16 ms per frame (real-time 60 FPS). For 720p→2K, latency is only 7 ms per frame (approximately 143 FPS), which is sufficient even for 120 Hz gaming. All 2K/4K clips are played by different users, collected after training, and remain strictly unseen during training and validation, mirroring the protocol used for our 1080p test set.

Table 3: Super-resolution performance of GameSR on high-resolution CS:GO clips (e.g., 720p→2K). All experiments were conducted on an NVIDIA A4000 GPU.

| Source → Target | Time (ms) | PSNR (dB) | SSIM | VMAF |
|---|---|---|---|---|
| 720p → 2K | 7.0 | 36.24 | 0.998 | 91 |
| 1080p → 4K | 14.3 | 39.25 | 0.999 | 93 |
| 720p → 4K | 15.3 | 34.83 | 0.999 | 88.13 |

Figure 5: e2e latency

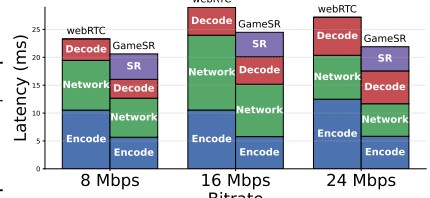

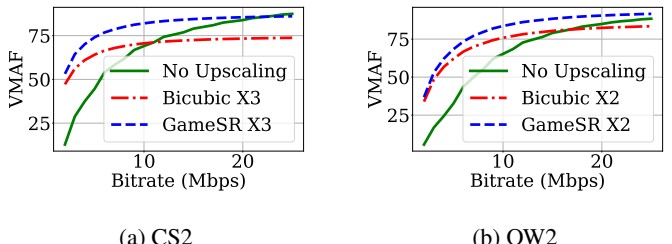

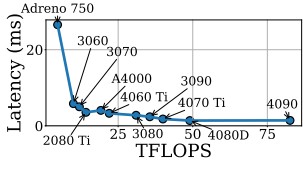

(a) CS2      (b) OW2      (c) Inference Latency vs TFLOPS

Figure 6: Performance of GameSR across bitrates (CS2, OW2) and compute capability (TFLOPS).

Despite being trained solely on 1080p content, GameSR can be directly applied to substantially higher-resolution inputs without fine-tuning, while preserving temporal stability, maintaining high perceptual quality, and sustaining low latency. This demonstrates that GameSR scales effectively to practical high-resolution cloud gaming scenarios.

**User Study.** To further assess perceptual quality, we conducted a user study using recorded gameplay sessions upscaled by GameSR. A total of **15 participants** took part, each watching **8 session recordings** across both CS2 and OW2. Among the participants, 36% were experienced gamers (Exp.), 36% were occasional gamers (Occ.), and 27% had little or no prior gaming experience (Not-gamer). After each viewing, participants rated the visual quality on a 5-point Mean Opinion Score (MOS) scale. The results (Table 4) demonstrate consistently high perceptual quality. GameSR achieved an average MOS of **4.73/5** for CS2 and **4.70/5** for OW2. Importantly, scores were consistent across all participant groups. These findings validate GameSR's ability to deliver high-quality perceptual results across diverse audiences, complementing our objective evaluation metrics.

**Inference Evaluation Across Heterogeneous Hardware.** We evaluate the scalability of GAMESR across a diverse set of client devices ranging from modern mobile SoCs to high-end desktop GPUs. Figure 6c reports the inference latency of GAMESR for 2× upscaling as a function of device TFLOPS. GAMESR-M (designed with mobile-friendly operations only) achieves real time performance on an Adreno 750 mobile GPU (4.7 TFLOPS), with 26.56 ms per 1080p frame for 2× and 14.55 ms per frame for 3×, which is within a 60 fps budget. On discrete GPUs, latency decreases roughly with available compute: mid range GPUs such as the RTX 4060 Ti, 4070 Ti, and A4000 sustain about 1.7 to 4.0 ms per frame for 2×, while high end cards like the RTX 3090 and 4090 reduce latency to about 2.3 ms and 1.3 ms, respectively. Overall, these results highlight the hardware scalability of GAMESR and its ability to meet real time budgets across a wide spectrum of client devices.

**Ablation Study.** To assess each component of GameSR, we ablated ConvLSTM, PixelUnshuffle, and Reparameterization, comparing inference time, memory, parameters, and quality (PSNR, SSIM, LPIPS). Results are shown in Table 7 (§A.6), which is moved to the appendix due to space constraints.. Removing ConvLSTM reduced parameters to 65K and inference to 3.05 ms, but quality dropped by ∼5 dB PSNR (37.99→32.99), showing the necessity of temporal modeling. Without PixelUnshuffle, PSNR peaked at 38.65 dB, but inference slowed to 13.13 ms and memory nearly doubled (1174 MiB), confirming its role in balancing fidelity and efficiency. Disabling reparameterization raised parameters by 54% (138K→298K) and inference to 6.29 ms with no quality gain.

Table 4: User study results (MOS) for GameSR across participant groups. Exp.=Experienced, Occ.=Occasional, No.=Non-gamers.

| Game | MOS | Exp. | Occ. | No. |
|------|-----|------|------|-----|
| CS2 | 4.73 | 4.62 | 4.75 | 4.84 |
| OW2 | 4.70 | 4.68 | 4.75 | 4.68 |

Table 5: Comparison of GameSR against IMDN and IMDN-G trained on CS2.

| Model | PSNR | SSIM | LPIPS |
|-------|------|------|-------|
| IMDN | 35.37 | 0.9988 | 0.0898 |
| IMDN-G | **38.17** | **0.9990** | **0.0835** |
| GameSR | 37.99 | 0.9990 | 0.0954 |

Overall, temporal modeling, feature restructuring, and reparameterization are all crucial for achieving real-time, high-quality performance under resource constraints.

## 4.3 END-TO-END CLOUD GAMING SYSTEM

While this paper focuses on the design and analysis of GameSR, we also implemented a complete cloud gaming testbed to validate its end-to-end feasibility. The system is built on aiortc/WebRTC and includes three neural components: (i) GameSR running on the client after video decode; (ii) a Complexity Analyzer, an MLP that predicts the average frame complexity of the next GOP from the previous two; and (iii) a Joint Optimizer, an RL-based rate controller on the server. Once per GOP, the Joint Optimizer takes the predicted complexity together with recent bandwidth and network statistics and outputs the game's rendering resolution and encoder bitrate. Video is encoded using low-latency H.264, sent over WebRTC, and all decoded frames are upscaled by GameSR on the client before display. As a baseline, we use WebRTC configured to always render and capture at 1080p, with stock Google Congestion Control and no client-side super-resolution; in all experiments, GameSR sits in the client decode–display path and processes every frame.

**Evaluation summary.** We deploy a gaming server and client connected via 1 Gbps Ethernet and stream 50 gaming sessions (25 CS2, 25 OW2) through aiortc/WebRTC under real-life latency traces from five geographic regions and two bandwidth regimes (30 Mbps and 8.5 Mbps) captured during a cloud gaming tournament. At 30 Mbps, both baseline WebRTC and our system maintain VMAF scores close to the target value of 90, but our framework uses only about 15 Mbps on average, i.e., up to 50% bandwidth savings compared to always rendering and encoding at 1080p. The adaptive render resolution plus GameSR also reduce client processing load: we observe up to 62% CPU and 41% GPU reductions relative to the fixed-1080p baseline. At 8.5 Mbps, where the link is bandwidth-constrained, our system improves the average VMAF by up to 33% across sessions.

To assess user experience, we conducted a subjective study with 15 participants and approximately 200 played sessions across CS2 and OW2 at both 30 Mbps and 8.5 Mbps. At 8.5 Mbps, our system improves the average MOS by up to 38% (OW2) and 34% (CS2) compared to baseline WebRTC. At 30 Mbps, it achieves comparable or slightly higher MOS while reducing bandwidth by about 50%, with MOS gains of 3.4% (OW2) and 4.6% (CS2).

Figure 5 breaks down end-to-end latency into encoding, network, decoding, and client processing for GT and GameSR at 8, 16, and 24 Mbps. Although GameSR adds about 4–5 ms of client-side processing per frame, the lower-resolution rendering and bitrate reduce encoder and network delay and decoding. Together with the rate–distortion curves in Fig. 6a and 6b, which show that GameSR reaches VMAF 80–90 at 30–60% lower bitrates than native 1080p across a wide range of encoder rates, this indicates that the quality gains from GameSR translate directly into end-to-end bandwidth and processing reductions in a realistic WebRTC pipeline. More details in AppendixA.8

## 5 CONCLUSION

We introduced GameSR, a fast and engine-agnostic super-resolution model for gaming. Unlike current upscalers, such as DLSS/FSR, GameSR requires no renderer data, enabling deployment in both modern and legacy games. Through efficient feature extraction, reparameterization, and lightweight temporal modeling, it achieves ∼4 ms inference time while preserving high quality. Objective and subjective experiments demonstrate that GameSR can save up to 60% of the bandwidth, and it consistently produces high perceived quality. Overall, GameSR offers a deployable path toward high-quality, low-cost, and real-time cloud gaming.

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

# A APPENDIX

This appendix supplements the main paper with extended descriptions of baseline super-resolution models, additional details of the GameSR architecture and training, and expanded evaluation results on quality and computational performance.

## A.1 DESCRIPTION OF IMAGE/VIDEO SR MODELS

Image and video SR has seen dramatic advancements in the last several years. The pursuit of real-time super-resolution has emerged as a significant research focus, with numerous approaches aimed at optimizing neural architectures for speed while maintaining quality (Ignatov et al., 2022; Li et al., 2022; Conde et al., 2023). Recent efforts have explored various optimization strategies, including network architecture refinement (Andrey Ignatov et al., 2021), model compression techniques, and innovative training approaches to reduce computational complexity, parameter count, and memory consumption.

Several notable architectures have made significant contributions toward real-time SR:

Information Multi-Distillation Network (**IMDN**) (Hui et al., 2019) introduces a lightweight architecture that efficiently extracts hierarchical features through cascaded blocks. The network's key innovation lies in its information distillation mechanism (IDM). Which progressively extracts and distills features at different scales. This approach enables the network to maintain high-quality outputs while significantly reducing computational overhead.

Laplacian Pyramid Super-Resolution Network (**LapSRN**) (Lai et al., 2017) implements a progressive upsampling strategy through a deep Laplacian pyramid structure. By stacking multiple upsampling layers, LapSRN achieves efficient resolution enhancement while maintaining control over computational complexity. The pyramid structure allows the network to reconstruct high-resolution images in a coarse-to-fine manner.

**LatticeNet** (Luo et al., 2020) introduces an innovative approach to parameter efficiency through its lattice block (LB) design. Inspired by lattice filter banks, the architecture combines residual blocks using a butterfly structure with attention mechanisms. This novel configuration achieves a remarkable 50% reduction in parameter count compared to traditional residual block-based models while maintaining comparable super-resolution quality.

**SwinIR** (Liang et al., 2021) introduces Transformer-based modeling to real-time super-resolution. Built on the Swin Transformer, it combines local self-attention with shifted windows to capture both short- and long-range dependencies efficiently. Its architecture integrates shallow convolutional features with deep features extracted via residual Swin Transformer blocks (RSTBs), enabling high-quality reconstruction with fewer parameters. SwinIR achieves state-of-the-art performance across multiple benchmarks.

However, despite these advances in lightweight architectures, meeting the stringent latency requirements of cloud gaming remains challenging. While these models successfully reduce computational complexity and memory usage, their architectures are primarily optimized for general efficiency rather than the specific speed requirements of real-time gaming applications.

**Video Super-Resolution (VSR)** extends these single-image approaches by incorporating temporal information from frame sequences. While single-image SR models focus purely on spatial en-

hancement, VSR processes either previous frames only (uni-directional) or both previous and future frames (bi-directional) to improve reconstruction quality (Fan et al., 2019; Li et al., 2020). However, several key limitations make existing VSR approaches unsuitable for real-time cloud gaming:

1. Computational Overhead: VSR models typically employ complex alignment modules, either explicit through optical flow (Dosovitskiy et al., 2015) or implicit via deformable convolutions (Shi et al., 2024). These alignment operations introduce significant computational costs, especially problematic for real-time processing.

2. Latency Requirements: Many VSR architectures process multiple frames simultaneously or require future frames, making them incompatible with cloud gaming's strict per-frame latency requirements.

3. Memory Constraints: State-of-the-art VSR models like SwinIR (Liang et al., 2021) use sophisticated architectures with multiple residual Swin Transformer blocks and self-attention mechanisms, requiring substantial memory to store temporal features.

### A.2 Extended GameSR Architecture Details

Extended details of the GameSR model design, omitted from the main paper, are provided here.

**Feature Extraction Block.** The feature learning stage is responsible for learning complex non-linear mappings between LR and HR representations. Our Feature Extraction Block (FEB) employs a multi-stage convolution sequence optimized for both training and inference phases. During training, each FEB processes features through three sequential operations, formally expressed as $F_{out} = F_{1\times1}^{compress}(F_{3\times3}(F_{1\times1}^{expand}(F_{in})))$, where an initial 1×1 convolution expands features from $C$ to $2C$ channels, followed by a core 3×3 convolution operating in this expanded feature space, and finally, a 1×1 convolution reduces the features back to $C$ channels.

To optimize inference performance, we leverage the reparameterization technique (Deng et al., 2023) to collapse these three convolutions into a single equivalent operation: $F_{out} = F_{3\times3}^{reparam}(F_{in})$. This transformation preserves the learned mapping while significantly reducing computational overhead during real-time processing. The reparameterization process combines the weights of all three convolutions as $W_{3\times3}^{reparam} = W_{1\times1}^{compress} * W_{3\times3} * W_{1\times1}^{expand}$, enabling efficient inference without compromising the model's learned capabilities.

The block's architecture is further refined through careful selection of activation and normalization components. We incorporate GeLU non-linearity, defined as $\text{GeLU}(x) = x \cdot \Phi(x)$ where $\Phi(x)$ is the cumulative distribution function of the standard normal distribution, providing smooth gradient flow during training. For normalization, we employ LayerNorm (Ba et al., 2016), expressed as $\text{LayerNorm}(x) = \gamma \cdot \frac{x-\mu}{\sqrt{\sigma^2+\epsilon}} + \beta$, where $\gamma$ and $\beta$ are learnable parameters, ensuring stable training behavior.

$$RB_1(F_1)_{f_1 \times r \times \frac{H}{s} \times \frac{W}{s}} = \max\left(0, W_{RB1} \cdot (F_1)_{f_1 \times \frac{H}{s} \times \frac{W}{s}} \right. \\ \left. + B_{RB1}\right) \tag{8}$$

Here, $W_{RB1}$ is a 1×1 convolution applied to the output of the feature expansion stage or the previous residual block. This layer expands the feature width from $f_1$ to $f_1 \cdot E$ (Lin et al., 2013), enabling richer representations. A non-linear activation follows to learn complex mappings before the next layer applies dimensionality reduction. The second layer of the block can be defined as:

$$RB_2(RB_1)_{f_1/r \times H \times W} = \max\left(0, \right. \\ W_{RB2} * (RB_1)_{f_1*E \times H \times W} \\ \left. + B_{RB2}\right) \tag{9}$$

Here, the $1 \times 1$ convolutional layer is applied to reduce the expanded features by a ratio of $r$. Once the channels are reduced, the final layer of the block can be defined as:

$$RB_3(RB_2)_{f_1 \times \frac{H}{s} \times \frac{W}{s}} = \text{LayerNorm}\Big(\text{Activation}\Big(\max\big(0, W_{RB3}$$
$$\cdot (RB_2)_{f_1 \times r \times \frac{H}{s} \times \frac{W}{s}} + B_{RB2}\big)\Big) + F_1\Big) \tag{10}$$

A final $3 \times 3$ convolution refines spatial features and restores the feature shape to $f_1 \times H \times W$. A residual connection adds the original input $F_1$ back to the output, preserving local details and maintaining consistent dimensions for the next residual block or ConvLSTM.

After extracting features through multiple FEBs, we employ a multi-level feature aggregation strategy to capture and combine representations at different abstraction levels. Unlike simple sequential processing, this approach allows the network to maintain and utilize both low-level details and high-level semantic information. Each successive FEB captures increasingly abstract features, with earlier blocks focusing on local patterns and textures, while deeper blocks capture more complex structural information.

To effectively combine these multi-scale representations, we employ an additive fusion strategy:

$$RB_{final} = \sum_{i=0}^{N} RB(i) \tag{11}$$

where $N$ represents the number of FEBs and $RB(i)$ denotes the output features from the $i$-th FEB. This additive combination offers several advantages:

1. Gradient Flow: The direct additive connections create shorter paths for gradient propagation during training, helping mitigate the vanishing gradient problem

2. Feature Reuse: Each subsequent layer can access and build upon features extracted at all previous levels, enabling more efficient feature utilization

4. Memory Efficiency: Unlike concatenation-based approaches that increase feature dimensionality, addition maintains a constant feature dimension while still preserving multi-level information

The empirical choice of $N$ FEBs balances model capacity with computational efficiency - too few blocks limit feature extraction capability, while too many increase computational overhead without proportional quality gains.

## A.3 ADDITIONAL TRAINING DETAILS

Beyond architectural considerations, the choice of loss and activation functions significantly impacts network performance and accuracy. For super-resolution tasks, three primary objective functions are commonly considered: Mean Squared Error (MSE), Mean Absolute Error (MAE), and Charbonnier loss. While MSE computes pixel-wise squared differences between generated and ground truth images, and MAE calculates absolute differences, we adopt the Charbonnier loss (Barron, 2017), which can be expressed as:

$$\text{Charbonnier Loss} = \mathbb{E}_{z,y \sim P_{\text{data}}(z,y)}\left[\rho(y - G(z))\right] \tag{12}$$

where $P(x) = \sqrt{x^2 + \epsilon^2}$. The Charbonnier loss functions as an adaptive combination of L1 and L2 losses, with its behavior governed by the parameter $\epsilon$. When the error exceeds $\epsilon$, it approximates L1 regularization; otherwise, it behaves more like L2 loss. Though L2 loss minimization typically maximizes PSNR, our empirical investigations revealed superior convergence characteristics with Charbonnier loss, leading to its adoption in our final implementation.

## A.4 VISUAL COMPARISON BETWEEN GAMESR, DLSS, AND FSR

Figure 7 shows side-by-side comparisons of upscaled frames from DLSS, FSR, and GameSR on Overwatch 2 sequences. The first, second, and third rows correspond to the Junktown, Esperança, and Nepal maps, respectively. We include OW2 here since it natively supports both DLSS and FSR, allowing direct visual comparison against GameSR.

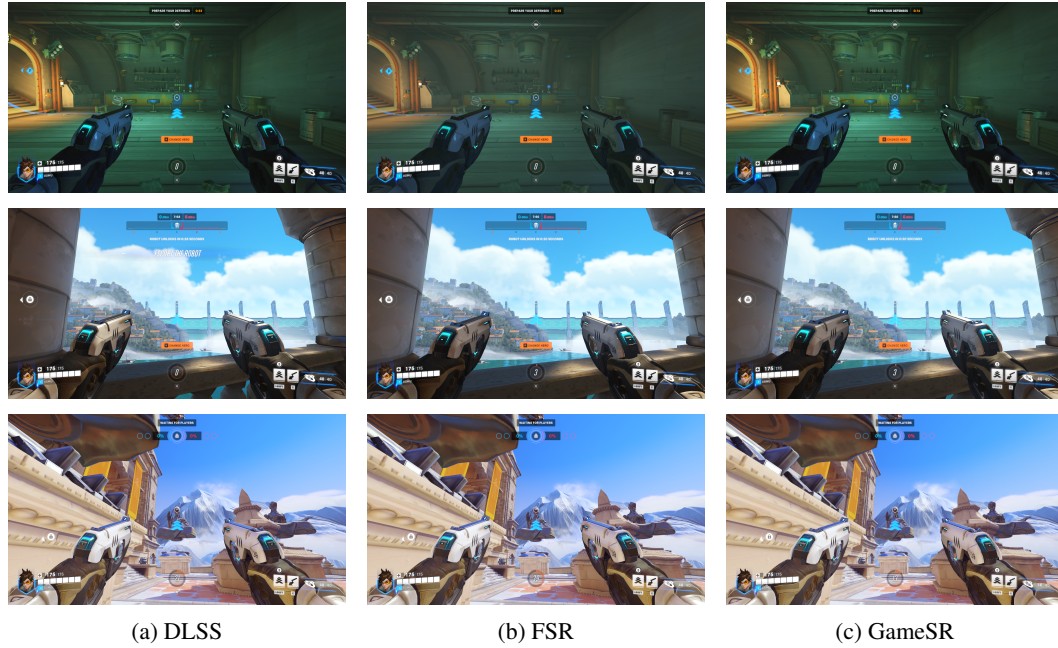

| (a) DLSS | (b) FSR | (c) GameSR |

Figure 7: Visualization of upsampling methods on Overwatch 2 frames: each row corresponds to a different map (Row 1: Junktown, Row 2: Esperança, Row 3: Nepal). Columns show DLSS, FSR, and GameSR, respectively.

Table 6: Quantitative comparison between state-of-the-art super-resolution models and GameSR at $3\times$ scaling on four popular games. Evaluated on an NVIDIA RTX A4000 GPU.

| Model | Inference (ms) | Counter-Strike 2 | | | Overwatch 2 | | | Team Fortress 2 | | | FIFA24 | | |
|---|---|---|---|---|---|---|---|---|---|---|---|---|---|
| | | PSNR | SSIM | LPIPS | PSNR | SSIM | LPIPS | PSNR | SSIM | LPIPS | PSNR | SSIM | LPIPS |
| SwinIR | 881.2 | 32.53 | 0.996 | **0.178** | 36.79 | **0.998** | **0.052** | 36.38 | **0.998** | **0.106** | **31.41** | **0.992** | 0.264 |
| LatticeNet | 70.5 | 32.25 | 0.996 | 0.189 | 36.42 | 0.9987 | 0.056 | 35.81 | 0.998 | 0.113 | 31.20 | 0.992 | 0.268 |
| EDSR | 76.6 | 32.26 | 0.996 | 0.188 | 36.32 | 0.9986 | 0.059 | 35.82 | 0.998 | 0.113 | 31.13 | 0.992 | 0.278 |
| IMDN | 55.2 | 32.29 | 0.996 | 0.186 | 36.40 | 0.9987 | 0.055 | 35.85 | 0.998 | 0.110 | 31.20 | 0.992 | 0.272 |
| GameSR | **4.09** | **35.46** | **0.996** | 0.180 | **36.99** | 0.998 | 0.059 | **38.10** | 0.998 | 0.111 | 31.23 | 0.992 | **0.253** |

## A.5 GameSR vs. SOTA Upscalers in Literature

In the main text (Section 4.2), we reported detailed comparisons for $2\times$ scaling (540p→1080p). For completeness, Table 6 presents results for $3\times$ scaling (360p→1080p). The trends mirror those observed at $2\times$: GameSR delivers quality on par with state-of-the-art SR models while being orders of magnitude faster.

## A.6 Ablation Study

To validate the contributions of each major component of GameSR, we performed an ablation study to evaluate the impact of ConvLSTM, PixelUnshuffle, and Reparameterization. Table 7 presents a comparison between different versions of GameSR, with each variant having one component removed. The comparison was based on inference time, memory usage, parameters, and quality metrics like PSNR, SSIM, and LPIPS.

The introduction of ConvLSTM enables temporal processing by utilizing information across multiple frames. The impact is significant: without ConvLSTM, the model's PSNR drops by approximately 5 dB (from 37.99 dB to 32.99 dB on CS2), with similar degradations in SSIM and LPIPS. While removing ConvLSTM reduces the parameter count to 65K and speeds up inference to 3.05 ms, the substantial quality loss demonstrates the critical importance of temporal information processing in our lightweight model.

Table 7: Ablation study of GameSR on CS2, showing the impact of ConvLSTM, PixelUnshuffle, and Reparameterization on efficiency and quality.

| Model | #Params (K) | Memory (MiB) | Inference (ms) | CS2 PSNR | SSIM | LPIPS |
|---|---|---|---|---|---|---|
| GameSR (No ConvLSTM) | **65** | **436** | **3.05** | 32.99 | 0.998 | 0.116 |
| GameSR (No PixelUnshuffle) | 125 | 1174 | 13.13 | 38.65 | 0.999 | 0.087 |
| GameSR (No Reparam.) | 298 | 608 | 6.29 | 37.99 | 0.998 | 0.095 |
| GameSR (Final Model) | 138 | 604 | 4.12 | 37.99 | 0.998 | 0.095 |

Table 8: Cross-game generalization of GameSR. Models trained on CS2, OW2, and a combined dataset are evaluated on both games.

| Test Sequence | Game data used for training | | | | | |
|---|---|---|---|---|---|---|
| | CS2 | | OW2 | | CS2+OW2 | |
| | PSNR | SSIM | PSNR | SSIM | PSNR | SSIM |
| CS2 | 37.80 | 0.999 | 34.63 | 0.999 | 37.53 | 0.999 |
| OW2 | 37.17 | 0.999 | 38.81 | 0.999 | 38.74 | 0.999 |

PixelUnshuffle proves essential for balancing quality and performance. Interestingly, the model without PixelUnshuffle achieves the highest quality metrics (PSNR: 38.65 dB on CS2), but at a severe efficiency cost. Inference time more than triples to 13.13 ms, and memory consumption nearly doubles to 1174 MiB, making it impractical for real-time applications. This trade-off highlights PixelUnshuffle's crucial role in preserving efficiency while maintaining strong quality.

The Reparameterization technique significantly improves model efficiency without compromising quality. Compared to the version without Reparameterization, our final model reduces parameters by 54% (298K $\to$ 138K) and improves inference time from 6.29 ms to 4.12 ms, while maintaining identical quality metrics. This demonstrates the effectiveness of reparameterization in optimizing deployment for resource-constrained environments.

### A.7 MODEL GENERALIZATION

To assess generalization, we evaluated models trained on CS2, OW2, and their combination across both games (Table 8). Models perform best in-domain (e.g., CS2-trained on CS2: PSNR 37.80, SSIM 0.999; OW2-trained on OW2: PSNR 38.82, SSIM 0.9996), but cross-game evaluations still yield competitive results, showing effective transfer. The combined CS2+OW2 model performs strongly on both, suggesting that shared motion and visual structures within the shooter genre improve robustness. These results demonstrate that GameSR adapts well across titles and benefits from multi-game training.

### A.8 END-TO-END CLOUD GAMING SYSTEM DETAILS

This appendix provides additional details about the end-to-end cloud gaming framework used to obtain the results in Sec. 4.3.

**Hardware and Network Setup.** The server is a Linux workstation (Intel Core i7-11800H, 16 GB RAM, RTX 3060, Ubuntu 22.04) and the client is a separate machine (Intel Xeon Gold 5220, 32 GB RAM, RTX A4000, Ubuntu 22.04), connected via a dedicated 1 Gbps Ethernet switch with no cross traffic.

**WebRTC Configuration.** The testbed is built on `aiortc` aio, a Python implementation of WebRTC. We use RTP over SRTP/UDP with stock Google Congestion Control (GCC) on the sender side. Video is encoded using H.264 via `libx264` with low-latency settings: `tune=zerolatency`, `bframes=0`, `preset=veryfast`, High profile, Level 4.2. The GOP size is aligned with the frame rate (key-int $\approx 1.5$ s; 90 frames at 60 fps and 45 at 30 fps), and we do not perform mid-GOP reconfiguration.

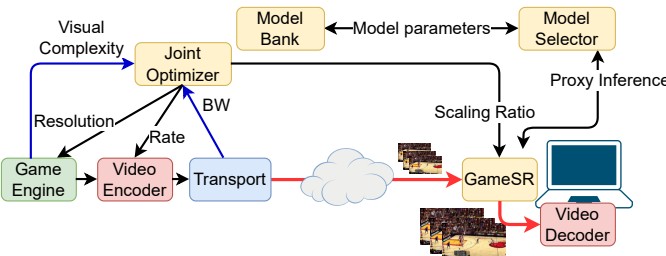

Figure 8: Overview of the proposed approach for jointly optimizing resources in cloud gaming.

**Network Traces and Emulation.** We emulate realistic wide-area conditions using latency traces from the CGCSDD cloud gaming dataset Alchalabi & Shirmohammadi (2021). The traces correspond to clients located in Toronto, Chicago, London, Brazil, and Singapore. For each gaming session, we select one trace and replay its per-packet RTT samples using Linux `tc netem`, configuring delay and jitter to follow the recorded values. We run experiments under two bandwidth regimes: 30 Mbps and 8.5 Mbps.

### A.8.1 JOINT OPTIMIZER

The Joint Optimizer (shown in figure8) balances server-side rendering and encoding overhead, transmission bandwidth, and client-side upsampling costs to optimize QoE under varying network and computational conditions. Its modular design enables integration with any underlying congestion or rate control algorithm. The Model Selector on the client-side performs proxy inference to find the model which satifies the latency budget on user hardware in real-time. The Model bank then from the server side based on the model selector inputs sends the weights for the right model.

**Reinforcement Learning.** At each GOP, our system must jointly choose a bitrate–resolution pair that balances server rendering/encoding cost, and network bandwidth, while maintaining high perceptual quality. We formulate bitrate–resolution selection as a Markov Decision Process and use *offline* reinforcement learning, trained on pre-collected traces with logged quality and resource statistics. We employed offline learning as quality is not available in real-time due to absense of ground truth during inference

The state at GOP $t$ is

$$s_t = [\phi_t, b_t^{\text{rec}}, u_t^{\text{gpu}}],$$

where $\phi_t$ is the predicted content complexity, $b_t^{\text{rec}}$ is the bitrate cap from the underlying congestion controller, and $u_t^{\text{gpu}}$ is the current GPU utilization. The action space is a discrete catalogue of bitrate–resolution pairs $(b, r)$ drawn from a bitrate ladder $\mathcal{B}$ and resolution set $\mathcal{R}$, with actions constrained by $b \leq b_t^{\text{rec}}$ at each GOP. The reward combines perceptual quality (VMAF) with penalties for server/client cost, and overload.

Because online exploration during gameplay, we adopt Discrete Batch-Constrained Q-learning (BCQ), which learns a $Q(s, a)$ function from offline transitions and constrains the policy to actions likely under the logged behavior policy, reducing out-of-distribution errors.

**Complexity Tiers and Predictor.** Our RL controller relies on a compact estimate of upcoming scene complexity. We adopt EVCA Amirpour et al. (2024) as our complexity metric, since it provides lightweight spatial and temporal scores from DCT-domain energy while remaining practical for online use. To avoid exposing raw, noisy values to RL, we cluster frame-level EVCA statistics from 25 full sessions using K-Means (after Z-score normalization) and obtain three tiers (Low, Mid, High). A small MLP then predicts the next GOP's tier from EVCA features of the current and previous two GOPs, achieving over 70% accuracy in 5-fold cross-validation.

### A.8.2 RESULTS

In the **30 Mbps** regime (Fig. 9), both baseline WebRTC and our framework achieve VMAF close to 90, but the Joint Optimizer with GameSR uses only about 15 Mbps on average, i.e., roughly **50% lower bitrate** than always streaming native 1080p. In a more constrained **8.5 Mbps** setting, our

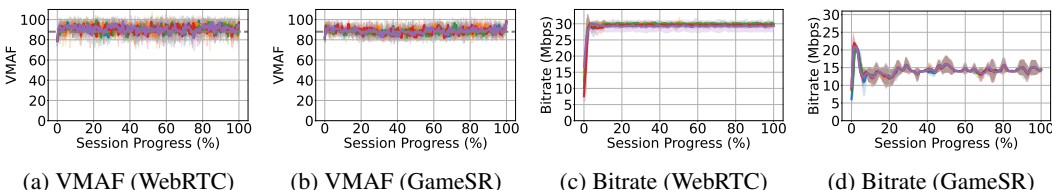

(a) VMAF (WebRTC)    (b) VMAF (GameSR)    (c) Bitrate (WebRTC)    (d) Bitrate (GameSR)

Figure 9: Cloud-gaming quality and bitrate behavior at a 30 Mbps link. Subfigures (a,b) show VMAF for baseline WebRTC and our system with GameSR, while (c,d) show the corresponding bitrate usage for the same sessions.

system raises the average VMAF by up to **33%** compared to WebRTC, with many sessions staying near 80 instead of dropping toward 60.

To assess perceptual quality, we conducted a subjective user study comparing our system against baseline WebRTC. We recruited **15 participants** (roughly balanced across experienced, occasional, and non-gamers). At **30 Mbps**, our system maintains essentially the same MOS as WebRTC while using about **50% less bandwidth**. At **8.5 Mbps**, it improves average MOS by about **34%** for CS2 and **38%** for OW2 relative to WebRTC, corresponding to a clear shift from "poor to fair" toward "good" perceived quality.

