# OpenReview forum: "GameSR: Real-Time Super-Resolution for Interactive Gaming"
_ICLR.cc/2026/Conference — Submitted to ICLR 2026_

### Official Review · Reviewer_AysD · 2025-10-19

**Soundness:** 3
**Presentation:** 2
**Contribution:** 2
**Rating:** 4
**Confidence:** 5

**Summary:**

The paper introduces GameSR, a lightweight and engine-independent super-resolution (SR) model targeting both cloud and local gaming scenarios. The proposed architecture combines PixelUnshuffle for feature rearrangement, re-parameterized convolutions for efficient inference, and a ConvLSTM module to capture temporal consistency across frames. The system operates as a post-processing step between video decoding and display, allowing real-time upscaling of rendered frames without modifying game engine code. Experiments were conducted on three commercial games (CS2, OW2, TF2) comparing GameSR with DLSS, FSR, and several academic SR models. The results suggest that GameSR achieves perceptual quality and runs in real time.

**Strengths:**

1. The topic is interesting and relevant to practical gaming applications.
2. The paper is evaluated on real commercial games instead of synthetic datasets, which adds realism.
3. Results look good in both quality and runtime, and the paper is overall easy to read.

**Weaknesses:**

1. The problem scope is confusing. The paper frames the work around cloud gaming, but it is not clear whether the target is a cloud setting or local post-processing on a client. The deployment point and constraints specific to cloud gaming are not well defined.
2. The technical novelty is limited. The model mainly combines several existing techniques, such as PixelUnshuffle, ConvLSTM, and re-parameterization, and the connections between them are not clearly motivated.
3. The claimed goals and the actual method do not really match. The paper emphasizes cloud deployment and lightweight design, but the model does not include any cloud-specific optimization, and the “lightweight” part is mostly limited to using PixelUnshuffle rather than any systematic efficiency strategy.

**Questions:**

1. The paper’s positioning is unclear. It claims to target cloud gaming, but the design does not include any cloud-specific optimization. At the same time, it suggests that the method also works locally, which essentially makes it a general SR approach without a clear focus.
2. If the goal is indeed cloud deployment, the input format deserves more discussion. For example, is the model aware of or adapted to video compression structures such as key frames and delta frames? How is the streaming or codec aspect considered?
3. The method appears to be a combination of existing techniques rather than a coherent new design. The components seem independent, with no evident co-design or joint optimization.
4. The lightweight design lacks substantive innovation. PixelUnshuffle and re-parameterization reduce computation for sure, but these are standard tricks, not specific design choices tailored for this work.
5. "In contrast, rendering at 540p and upscaling to 1080p using GameSR lowers utilization to ∼82%, reflecting reduced shading cost and lightweight inference." is not convincing. The reduction likely reflects idle cycles or a memory-bound condition rather than truly lighter computation.
6. The evaluation uses three FPS games that are visually and structurally similar, which weakens the claim of generalization.
7. The paper lacks comparison with the most recent game-oriented SR methods.

---

> ### Author Response · Authors · 2025-11-19
>
> We appreciate the reviewers’ constructive feedback. We have addressed the main concerns as follows:
>
> 1. **End-to-end cloud gaming system (Sec. 4.3).**
>     We now briefly describe our complete WebRTC-based system (RL-based rate controller \+ GameSR after H.264 decode). We also summarize the end-to-end results from over 50 CS2/OW2 gaming sessions under realistic latency and bandwidth conditions, including bandwidth/CPU/GPU savings and MOS gains.
>
> 2. **Expanded evaluation and scalability (Secs. 4.2, Table 3, Fig. 6(c)).**
>    We add 2K/4K experiments, heterogeneous hardware results from mobile GPUs (Adreno 750 / Samsung S24) to RTX GPUs, and a new game (FIFA24), showing that GameSR generalizes beyond first-person shooter games, scales to high resolutions, and supports various devices.
>
> 3. **Positioning and engine-agnostic design (Secs. 3.1).**
>    We clarify that GameSR is an engine- and codec-agnostic post-render, post-decoder upscaler for both local rendering and cloud gaming, complementing DLSS/FSR by working on legacy, closed-source, and multi-engine titles where render buffers are unavailable.
>
> 4. **Relation to prior edge / video SR methods (Sec. 2).**
>     We extend related work to discuss edge-SR, engine-integrated SR (SRPO, RDG, FuseSR, MNSS, Deep Fourier), and recurrent VSR (RLSP, MRVSR, SSL), explaining why their engine coupling, fixed 4X scale, and runtime make them complementary rather than direct baselines for our 2X/3X, 1080p, post-decoder setting.
>
> 5. **Metrics, perceptual quality, and ablations (Secs. 4.1, 4.2, App. A.6).**
>     We clarify how PSNR/SSIM/VMAF/LPIPS are computed, connect them to our 15-participant user study, and highlight existing ablations of ConvLSTM, PixelUnshuffle, and reparameterization, as well as our updated discussion of GPU utilization and frame-wise operation.
>
> More details about the revisions are provided below:
>
> * **Positioning of GameSR:** We clarify the positioning of GameSR in the revision. Our goal is to support **both local rendering and cloud gaming**, but always in an **engine-agnostic** way: GameSR runs purely on RGB frames and does not require access to G-buffers, motion vectors, depth, or vendor-specific hardware features (e.g., DLSS on RTX with dedicated integration). This makes it applicable to a much broader set of titles than current industrial upscalers: it can be used with legacy games such as TF2 on the original Source engine, with engines that do not expose the rendering buffers or motion/depth signals that modern DLSS/FSR plugins assume, and with cloud-gaming pipelines where only decoded video is available and transmitting extra render data would be too costly. Our **main comparison focus** is therefore on widely deployed game upsamplers (DLSS/FSR) because they represent the dominant practical baselines for high-quality gaming SR today; GameSR is designed to *complement* these methods by offering a vendor- and engine-independent alternative that works on both local PCs/consoles and cloud clients, including for titles and platforms where engine-integrated solutions are not feasible.
>
>
> * **On the practicality of GameSR and the advantage of an engine-free design:** We agree that engine-level signals (G-buffers, motion, depth) are very powerful and that for AAA titles with existing DLSS/FSR-style integrations the quality–integration trade-off is often acceptable. GameSR, however, is explicitly targeted at cloud gaming and legacy or multi-engine catalogues, where clients only see encoded/decoded RGB frames and engine buffers are either inaccessible (closed-source, older engines) or too costly to transmit at scale. In these settings, just integrating it into the engine is not realistic, so an engine-agnostic post-decode upscaler is required.

---

> > ### Author Response · Authors · 2025-11-19
> >
> > * **Comparisons against other works mentioned by the reviewers:**
> >
> >   * Our setting is fundamentally different from **GameStreamSR**, so we do not view it as a directly comparable baseline. GameStreamSR is an engine-dependent, depth-guided RoI upscaler: it applies a heavy SR network (instantiated with existing models such as EDSR) only to a small depth-defined region, and upscales the rest of the frame with bilinear interpolation. This design is tailored to mobile cloud gaming with access to render buffers and control over the engine, but leaves large portions of the image at bilinear quality (including HUD, distant targets, and text) and cannot be used in generic cloud-gaming clients that only see encoded/decoded RGB frames. In contrast, GameSR is a lightweight, full-frame post-render, post-decoder upscaler that operates purely on RGB video and can be deployed on both local and cloud clients, including legacy and closed-source titles where engine data are unavailable.
> >
> >   * **edge-SR** is built around very shallow (often 1–3 layer) single-image models, designed as drop-in replacements for classic upscalers on generic natural images. Modern 3D game frames are far more complex (high motion, fine geometry, HUDs, compression artifacts), and even **deeper** state-of-the-art SR baselines in our experiments struggle to fully recover accurate details. Given this, we do not expect a 1–3 layer image-only architecture to be competitive in our high-resolution, temporally demanding games.
> >
> >   * **SRPO** (ECCV’22), **RDG** (CVPR’25), **FuseSR**, **MNSS**, and **the Deep Fourier-based arbitrary-scale SR** method all target a different operating point than GameSR: they are tightly integrated into the real-time renderer and rely on internal engine signals such as high-resolution G-buffers, motion vectors, depth, and other per-pixel attributes. This requires modifying or instrumenting the game engine and is mainly practical when one controls the renderer (e.g., in-house AAA titles). By contrast, GameSR is deliberately designed as a **post-render, post-decoder** upscaler that operates solely on RGB frames: it can run (i) after video decode in a cloud-gaming client, and (ii) after a lower-resolution render target is resolved locally (e.g., 540p/720p→1080p) without any access to engine buffers or vendor-specific hardware. This makes it applicable to legacy and closed-source games, engines without forward-rendering data exposed, and generic streaming platforms where only decoded video is available. Moreover, these engine-coupled methods are typically evaluated in controlled rendering demos or author-created scenes rather than on long, recorded gameplay streams under encoding and network constraints as in our experiments. For these reasons, we view them as complementary engine-integrated approaches and do not treat them as directly comparable baselines for the post-render/post-decoder deployment scenario that GameSR targets.
> >
> >   * Regarding *Efficient Video Super-Resolution through Recurrent Latent Space Propagation* (**RLSP**), *Stable Long-Term Recurrent Video Super-Resolution* (**MRVSR**), and ***Structured Sparsity Learning for Efficient Video Super-Resolution***, we agree that they are important recurrent VSR baselines, but they target a very different domain than ours. All three are fixed x4 models evaluated on small LR inputs (typically 180×320→720p), whereas in gaming and modern upscalers such as DLSS/FSR the practical setting for 1080p is closer to 2x (e.g., 960×540→1080p or 720p→1080p); pushing to 4x from very low LR (e.g., 270p→1080p) is generally avoided because quality and object visibility degrade noticeably in games. A single fixed 4x scale also offers limited flexibility for any intelligent rate controller that needs to adapt rendering resolution to network congestion (e.g., switching between 360p/540p/720p with corresponding 2x/3x upscaling factors), which a 4x-only design cannot support. In addition, the reported runtimes for these models are on the order of tens of milliseconds per frame at 180×320 on high-end GPUs (TITAN RTX, TITAN Xp, V100), whereas GameSR achieves 2x/3x upscaling for full-HD gaming streams (540p/720p→1080p) in about 4–5 ms per frame on a similar GPU in terms of power with only 138K parameters. Adapting these architectures to our 1080p, 2x/3x, 60 fps setting would require substantial redesign and still provide less control to an adaptive rate controller, so we treat them as complementary general VSR work rather than directly comparable baselines for the deployment we target.

---

> > > ### Author Response · Authors · 2025-11-19
> > >
> > > * **Expanded Evaluation:**
> > >
> > >   * **E2E Cloud Gaming System:** Our original submission focused on the SR component in isolation, but we have, in fact, implemented and evaluated a complete end-to-end cloud gaming system built on aiortc/WebRTC. The system includes an RL-based Joint Optimizer with a Complexity Analyzer on the server that selects the rendering resolution and encoder rate once per group of pictures (GOP), and GameSR running on the client-side after H.264 decode. In the revised paper, we add Sec. 4.3, “End-to-End Cloud Gaming System”, where we summarize this design and report results over 50 gaming sessions of CS2 and OW2 under realistic latency traces and two bandwidth regimes (30 Mbps and 8.5 Mbps). Compared to baseline WebRTC, our framework achieves savings of up to 50%, 62%, and 41% in bandwidth, CPU, and GPU requirements, respectively, while maintaining VMAF close to 90 and significantly improving quality at 8.5 Mbps. A separate subjective user study with 15 participants and approximately 200 played sessions further shows large MOS gains of up to 38% at 8.5 Mbps, and at 30 Mbps, our system maintains similar perceived quality while reducing bandwidth by about 50%, with MOS improving by 3.4% (OW2) and 4.6% (CS2) relative to baseline WebRTC. All these measurements include rendering, capture, encoding, network transmission, decoding, and GameSR upscaling; this indicates that the complete pipeline with GameSR integrated can satisfy real-time cloud gaming constraints in practice.
> > >
> > >   * **Dataset Generalization & Scalability:** We appreciate the concern about game and genre diversity and agree that this is an important dimension. In this work, we chose to focus on three shooter titles and to train GameSR per game, similar to how DLSS and FSR are typically tuned per title in practice. We also understand that CS2, OW2, and TF2 may feel similar, since they are all shooters. However, from a rendering and content perspective they are quite different: CS2 is a realistic tactical FPS on the Source 2 engine with high-contrast lighting and fine geometric detail; Overwatch 2 is a fast-paced hero shooter on Blizzard’s Overwatch engine with highly stylized characters and dense particle effects; and TF2 is an older class-based shooter on the original Source engine with flatter shading and heavy HUD elements. In the revision, we also add a fourth title, FIFA, a third-person sports game with long-range camera views, large uniform textures (pitch/grass), and different motion patterns (passes, shots, camera pans), and report its results in Sec .\~4.2 (Table\~2). As shown there, FIFA follows the same trend as our other games in terms of quality and efficiency, illustrating that GameSR extends naturally beyond FPS titles and that adding a new game requires only training on that game’s footage. In addition, these four titles span engines with very different support for industrial upscalers: recent engines where DLSS/FSR-style plugins are available, engines that only support FSR-like spatial upscalers because the forward-rendering data required by DLSS-style methods is not exposed, and a legacy engine (TF2) where neither DLSS nor modern FSR variants can be deployed. One might argue that heuristic upsamplers such as bicubic, bilinear, or Lanczos are sufficient for such older games, but our results in Table\~2 show that even for TF2, the bicubic baseline is consistently weaker than GameSR across all scales, both in reconstruction quality and perceptual metrics.
> > >
> > >   * **Inclusion of 2K & 4K results:** We focus on 1080p because, according to the Steam Hardware & Software Survey (October 2025), it is still the dominant gaming resolution (≈53% of players), while 1440p and 4K account for ≈20% and ≈5%, respectively, so 1080p remains the most practical operating point for real-time cloud gaming and streaming. To directly address the concern about higher resolutions, we have added 2K and 4K experiments in Sec. 4.2 (“Scalability and Generality to High-Resolution Upscaling,” Table 3): GameSR reaches up to 39.25 dB PSNR, SSIM \> 0.999, and VMAF \> 93 for 1080p→4K while keeping inference \< 16 ms per frame (real-time 60 FPS), and only 7 ms per frame for 720p→2K (≈143 FPS, suitable for 120 Hz). These results are obtained on unseen 2K/4K clips collected after training, showing that a model trained solely at 1080p **generalizes and scales** to substantially higher resolutions without fine-tuning, while maintaining high perceptual quality and low latency for practical high-resolution cloud gaming scenarios.

---

> ### Author Response · Authors · 2025-11-19
>
> * **Evaluation Metric:** We apologize for the lack of clarity and will make the metric pipeline explicit. For all objective metrics (PSNR, SSIM, VMAF, LPIPS), the **high-resolution reference** is always the native game capture at full resolution (1080p in the main experiments, and 2K/4K in Sec. 4.2). The **low-resolution inputs** to GameSR are generated by downscaling these native captures: for 2x experiments we use 540p→1080p, for 3x we use 360p→1080p (and similarly 720p→2K, 1080p→4K, 720p→4K in the scalability experiments); in the cloud-gaming RD experiments, these LR sequences are additionally encoded and decoded to mimic a real streaming pipeline before being fed to GameSR. We then compute frame-wise PSNR/SSIM between the GameSR output and the native HR reference and report averages over each clip and game. VMAF is computed using the official Netflix implementation on the HR reference and the upscaled sequence, and LPIPS is computed using the standard PyTorch implementation; both have been widely shown to correlate well with human perception. Finally, our user study with 15 participants (Table 4\) yields high MOS scores (≈4.7/5) and consistent ratings across gamer/non-gamer groups, which supports that our VMAF/LPIPS trends indeed reflect perceived visual quality.
>
> * **Hardware Scalability:** We apologize for the confusion and will clarify the intended hardware targets in the revision. GameSR is designed as a client-side upscaler for commodity cloud-gaming devices ranging from smartphones to desktop GPUs, not only desktops. To make this explicit, we have added Sec. 4.2 (“Inference Evaluation Across Heterogeneous Hardware,” Fig. 6(c), where we benchmark both the original GameSR and a mobile-oriented variant, GameSR-M, across devices from an Adreno 750–class Samsung S24 mobile GPU (4.7 TFLOPS) up to RTX 40-series desktop GPUs. On the Adreno 750, GameSR-M already achieves real-time performance with 26.56 ms (X2) and 14.55 ms (X3) per 1080p frame, demonstrating feasibility on modern handhelds where NPUs/GPUs are typically underutilized during cloud gaming. On discrete GPUs, latency scales roughly inversely with compute: mid-range cards (RTX 4060 Ti / 4070 Ti / A4000) sustain 1.7–4.0 ms per frame, while high-end RTX 3090/4090 reach 2.3 ms and 1.3 ms, respectively. These results show that our design is explicitly hardware-scalable and can even run on resource-constrained devices.
>
> * **Clarifications:**
>   * **GPU utilization:** In the configuration used for Fig. 4(c), the measured frame rates were approximately 123 FPS (native 1080p), 140 FPS (DLSS), 145 FPS (FSR), and about 125 FPS for GameSR. Thus, GameSR achieves essentially the same FPS as native while using only ≈82% GPU utilization, whereas native/DLSS/FSR run at similar or higher FPS but keep the GPU near 99–100% utilization. Both DLSS and FSR internally render at reduced resolution and then upscale to display resolution, and in typical usage the recovered shading budget is spent on higher FPS rather than reducing GPU load; this is consistent with our original text that “*DLSS/FSR are intended to boost framerate rather than reduce computation.*” Our point in Fig. 4(c) is therefore not that DLSS/FSR are inefficient or that high GPU utilization is inherently negative, but that the GameSR configuration can match native frame rate while leaving more GPU headroom. In a cloud-gaming deployment, where rendering is done on the server and upscaling is done on the client, this headroom can be converted into lower server-side rendering cost and reduced streaming bitrate by rendering at lower resolution and adding only a small SR overhead on the client. We have updated the main text around Fig. 4(c) to explicitly report these FPS numbers together with the utilization trends.
>
>   * **Ablation study:** We already provide an ablation in Sec. 4.2 and App. A.6 (Table 7), where we remove ConvLSTM, PixelUnshuffle, and reparameterization and jointly measure PSNR/SSIM/LPIPS, latency, parameters, and memory. For example, removing ConvLSTM reduces latency to 3.05 ms and parameters to 65K but degrades PSNR by about 5 dB (37.99 → 32.99), and the ConvLSTM hidden state is kept locally on the client and never transmitted, so it adds no bandwidth overhead. We will surface these key numbers more prominently in the main paper.
>
>   * **Key/delta frames:** In our cloud-gaming setting, the client receives a live stream of decoded RGB frames, so GameSR is designed to operate per frame and is agnostic to codec internals such as I/P/B frame types. Given its low latency (approximately 3–6 ms per 1080p frame on mid-range GPUs), there is no need to upscale only keyframes and interpolate the rest, a strategy that is more suitable for offline VOD pipelines than for interactive, high-motion gaming.

---

### Official Review · Reviewer_CgRw · 2025-10-25

**Soundness:** 3
**Presentation:** 3
**Contribution:** 2
**Rating:** 4
**Confidence:** 5

**Summary:**

The paper proposes GameSR, a neural real-time super-resolution framework for video gaming. Compared to existing industry gaming SR frameworks like DLSS or FSR, the proposed GameSR network does not rely on access to engine-level data, so it does not require game developers to explicitly integrate with game engines and is more suitable for cloud-gaming settings. It leverages pixel shuffling for a lightweight feature learning and ConvLSTM for temporal learning. Results shows that the upsampling quality is on par with industry SR frameworks while the runtime performance is significantly faster than existing image/video SR networks.

**Strengths:**

1. Engine-data-free pipeline. Without relying on rendering engine data, the pipeline can be more broadly applicable across all kinds of games and cloud gaming scenarios. Especially for cloud gaming scenarios, it can greatly save the network bandwidth.
2. The proposed method achieves better runtime performance and on-par quality with SOTA methods.

**Weaknesses:**

## Major weaknesses
1. Missing citations. The real-time SR problem for gaming has been an important research direction in computer graphics literature, but the citation to several important works are missing here:
```
@inproceedings{zhong2023fusesr,
  title={Fusesr: Super resolution for real-time rendering through efficient multi-resolution fusion},
  author={Zhong, Zhihua and Zhu, Jingsen and Dai, Yuxin and Zheng, Chuankun and Chen, Guanlin and Huo, Yuchi and Bao, Hujun and Wang, Rui},
  booktitle={SIGGRAPH Asia 2023 Conference Papers},
  pages={1--10},
  year={2023}
}
@article{yang2023mnss,
  title={Mnss: Neural supersampling framework for real-time rendering on mobile devices},
  author={Yang, Sipeng and Zhao, Yunlu and Luo, Yuzhe and Wang, He and Sun, Hongyu and Li, Chen and Cai, Binghuang and Jin, Xiaogang},
  journal={IEEE transactions on visualization and computer graphics},
  volume={30},
  number={7},
  pages={4271--4284},
  year={2023},
  publisher={IEEE}
}
@inproceedings{zhang2024deep,
  title={Deep Fourier-based arbitrary-scale super-resolution for real-time rendering},
  author={Zhang, Haonan and Guo, Jie and Zhang, Jiawei and Qin, Haoyu and Feng, Zesen and Yang, Ming and Guo, Yanwen},
  booktitle={ACM SIGGRAPH 2024 Conference Papers},
  pages={1--11},
  year={2024}
}
```
2. Missing baselines. The paper only provides a limited comparison to industry SR in Tab 1, while in Tab 2 all baselines are not designated for real-time gaming but traditional image/video SR methods, which is unfair. The paper should also include a comprehensive comparison to real-time SR papers, such as Neural Supersampling (NSRR), FuseSR, Deep Fourier SR, Mob-FGSR, etc.
3. GPU utilization. The paper claims a lower GPU utilization rate to be an advantage over the industry's fully-utilized methods, which does not make sense. In the gaming industry, low GPU utilization rates typically mean a poor optimization in rendering, since game engines are supposed to fully utilize the GPU computation for a maximum frame rate boost. A low GPU utilization rate may indicate a performance bottleneck in the rendering pipeline, blocking the GPU from working fully, which is not desired.
4. Temporal consistency. The paper only evaluates image-based metrics (e.g., PSNR), but does not evaluate the temporal consistency between frames, which is also crucial for real-time gaming. In particular, even if each individual frame may have a high PSNR compared to the corresponding ground truth frames, when playing them together as a video, significant flickering artifacts may still exist. Result videos can be a convincing experimental result, but the author does not include them as supplementary material.

## Minor
1. Typo in Fig 2: The channels of the features after PixelShuffle in the ConvLSTM row show "H/SxW/Sx3(S*2)^2", is this a typo? Should it be "H/SxW/Sx3(S^2)^2"?
2. Higher resolution experiments. The paper mainly uses 1080P as the target high resolution in the experiments. However, in modern gaming, 2K and even 4K are becoming more and more common. Existing papers (e.g. FuseSR) already include experiments in 4K settings, it would be great if this paper could also include this to see the method's potential in modern ultra-high resolution gaming.

**Questions:**

1. Please address the questions raised in the "Weakness" section.
2. Discussion on whether or not to use engine-level data. The paper emphasizes that not using engine-level data is an advantage over existing methods. I can't say I disagree with this claim, but I think it is worth discussing. In practice, engine-level data such as G-buffers are widely used in industry, as they provide rich scene information that can greatly enhance SR quality with relatively modest integration effort. For AAA games, this integration is typically acceptable and even facilitated by official plugins (e.g., NVIDIA has official DLSS plugin on Unreal Engine). In contrast, non-AAA titles are often less photo-realistic and may not rely heavily on SR due to lower rendering demands. Therefore, the trade-off between integration effort and SR quality might be smaller than the paper suggests. That said, I agree that in cloud-based rendering or streaming scenarios, avoiding engine-level data is indeed advantageous, as it reduces network overhead and improves scalability.

---

> ### Author Response · Authors · 2025-11-19
>
> We appreciate the reviewers’ constructive feedback. We have addressed the main concerns as follows:
>
> 1. **End-to-end cloud gaming system (Sec. 4.3).**
>     We now briefly describe our complete WebRTC-based system (RL-based rate controller \+ GameSR after H.264 decode). We also summarize the end-to-end results from over 50 CS2/OW2 gaming sessions under realistic latency and bandwidth conditions, including bandwidth/CPU/GPU savings and MOS gains.
>
> 2. **Expanded evaluation and scalability (Secs. 4.2, Table 3, Fig. 6(c)).**
>    We add 2K/4K experiments, heterogeneous hardware results from mobile GPUs (Adreno 750 / Samsung S24) to RTX GPUs, and a new game (FIFA24), showing that GameSR generalizes beyond first-person shooter games, scales to high resolutions, and supports various devices.
>
> 3. **Positioning and engine-agnostic design (Secs. 3.1).**
>    We clarify that GameSR is an engine- and codec-agnostic post-render, post-decoder upscaler for both local rendering and cloud gaming, complementing DLSS/FSR by working on legacy, closed-source, and multi-engine titles where render buffers are unavailable.
>
> 4. **Relation to prior edge / video SR methods (Sec. 2).**
>     We extend related work to discuss edge-SR, engine-integrated SR (SRPO, RDG, FuseSR, MNSS, Deep Fourier), and recurrent VSR (RLSP, MRVSR, SSL), explaining why their engine coupling, fixed 4X scale, and runtime make them complementary rather than direct baselines for our 2X/3X, 1080p, post-decoder setting.
>
> 5. **Metrics, perceptual quality, and ablations (Secs. 4.1, 4.2, App. A.6).**
>     We clarify how PSNR/SSIM/VMAF/LPIPS are computed, connect them to our 15-participant user study, and highlight existing ablations of ConvLSTM, PixelUnshuffle, and reparameterization, as well as our updated discussion of GPU utilization and frame-wise operation.
>
> More details about the revisions are provided below:
>
> * **Positioning of GameSR:** We clarify the positioning of GameSR in the revision. Our goal is to support **both local rendering and cloud gaming**, but always in an **engine-agnostic** way: GameSR runs purely on RGB frames and does not require access to G-buffers, motion vectors, depth, or vendor-specific hardware features (e.g., DLSS on RTX with dedicated integration). This makes it applicable to a much broader set of titles than current industrial upscalers: it can be used with legacy games such as TF2 on the original Source engine, with engines that do not expose the rendering buffers or motion/depth signals that modern DLSS/FSR plugins assume, and with cloud-gaming pipelines where only decoded video is available and transmitting extra render data would be too costly. Our **main comparison focus** is therefore on widely deployed game upsamplers (DLSS/FSR) because they represent the dominant practical baselines for high-quality gaming SR today; GameSR is designed to *complement* these methods by offering a vendor- and engine-independent alternative that works on both local PCs/consoles and cloud clients, including for titles and platforms where engine-integrated solutions are not feasible.
>
>
> * **On the practicality of GameSR and the advantage of an engine-free design:** We agree that engine-level signals (G-buffers, motion, depth) are very powerful and that for AAA titles with existing DLSS/FSR-style integrations the quality–integration trade-off is often acceptable. GameSR, however, is explicitly targeted at cloud gaming and legacy or multi-engine catalogues, where clients only see encoded/decoded RGB frames and engine buffers are either inaccessible (closed-source, older engines) or too costly to transmit at scale. In these settings, just integrating it into the engine is not realistic, so an engine-agnostic post-decode upscaler is required.

---

> > ### Author Response · Authors · 2025-11-19
> >
> > * **Comparisons against other works mentioned by the reviewers:**
> >
> >   * Our setting is fundamentally different from **GameStreamSR**, so we do not view it as a directly comparable baseline. GameStreamSR is an engine-dependent, depth-guided RoI upscaler: it applies a heavy SR network (instantiated with existing models such as EDSR) only to a small depth-defined region, and upscales the rest of the frame with bilinear interpolation. This design is tailored to mobile cloud gaming with access to render buffers and control over the engine, but leaves large portions of the image at bilinear quality (including HUD, distant targets, and text) and cannot be used in generic cloud-gaming clients that only see encoded/decoded RGB frames. In contrast, GameSR is a lightweight, full-frame post-render, post-decoder upscaler that operates purely on RGB video and can be deployed on both local and cloud clients, including legacy and closed-source titles where engine data are unavailable.
> >
> >   * **edge-SR** is built around very shallow (often 1–3 layer) single-image models, designed as drop-in replacements for classic upscalers on generic natural images. Modern 3D game frames are far more complex (high motion, fine geometry, HUDs, compression artifacts), and even **deeper** state-of-the-art SR baselines in our experiments struggle to fully recover accurate details. Given this, we do not expect a 1–3 layer image-only architecture to be competitive in our high-resolution, temporally demanding games.
> >
> >   * **SRPO** (ECCV’22), **RDG** (CVPR’25), **FuseSR**, **MNSS**, and **the Deep Fourier-based arbitrary-scale SR** method all target a different operating point than GameSR: they are tightly integrated into the real-time renderer and rely on internal engine signals such as high-resolution G-buffers, motion vectors, depth, and other per-pixel attributes. This requires modifying or instrumenting the game engine and is mainly practical when one controls the renderer (e.g., in-house AAA titles). By contrast, GameSR is deliberately designed as a **post-render, post-decoder** upscaler that operates solely on RGB frames: it can run (i) after video decode in a cloud-gaming client, and (ii) after a lower-resolution render target is resolved locally (e.g., 540p/720p→1080p) without any access to engine buffers or vendor-specific hardware. This makes it applicable to legacy and closed-source games, engines without forward-rendering data exposed, and generic streaming platforms where only decoded video is available. Moreover, these engine-coupled methods are typically evaluated in controlled rendering demos or author-created scenes rather than on long, recorded gameplay streams under encoding and network constraints as in our experiments. For these reasons, we view them as complementary engine-integrated approaches and do not treat them as directly comparable baselines for the post-render/post-decoder deployment scenario that GameSR targets.
> >
> >   * Regarding *Efficient Video Super-Resolution through Recurrent Latent Space Propagation* (**RLSP**), *Stable Long-Term Recurrent Video Super-Resolution* (**MRVSR**), and ***Structured Sparsity Learning for Efficient Video Super-Resolution***, we agree that they are important recurrent VSR baselines, but they target a very different domain than ours. All three are fixed x4 models evaluated on small LR inputs (typically 180×320→720p), whereas in gaming and modern upscalers such as DLSS/FSR the practical setting for 1080p is closer to 2x (e.g., 960×540→1080p or 720p→1080p); pushing to 4x from very low LR (e.g., 270p→1080p) is generally avoided because quality and object visibility degrade noticeably in games. A single fixed 4x scale also offers limited flexibility for any intelligent rate controller that needs to adapt rendering resolution to network congestion (e.g., switching between 360p/540p/720p with corresponding 2x/3x upscaling factors), which a 4x-only design cannot support. In addition, the reported runtimes for these models are on the order of tens of milliseconds per frame at 180×320 on high-end GPUs (TITAN RTX, TITAN Xp, V100), whereas GameSR achieves 2x/3x upscaling for full-HD gaming streams (540p/720p→1080p) in about 4–5 ms per frame on a similar GPU in terms of power with only 138K parameters. Adapting these architectures to our 1080p, 2x/3x, 60 fps setting would require substantial redesign and still provide less control to an adaptive rate controller, so we treat them as complementary general VSR work rather than directly comparable baselines for the deployment we target.

---

> > > ### Author Response · Authors · 2025-11-19
> > >
> > > * **Expanded Evaluation:**
> > >
> > >   * **E2E Cloud Gaming System:** Our original submission focused on the SR component in isolation, but we have, in fact, implemented and evaluated a complete end-to-end cloud gaming system built on aiortc/WebRTC. The system includes an RL-based Joint Optimizer with a Complexity Analyzer on the server that selects the rendering resolution and encoder rate once per group of pictures (GOP), and GameSR running on the client-side after H.264 decode. In the revised paper, we add Sec. 4.3, “End-to-End Cloud Gaming System”, where we summarize this design and report results over 50 gaming sessions of CS2 and OW2 under realistic latency traces and two bandwidth regimes (30 Mbps and 8.5 Mbps). Compared to baseline WebRTC, our framework achieves savings of up to 50%, 62%, and 41% in bandwidth, CPU, and GPU requirements, respectively, while maintaining VMAF close to 90 and significantly improving quality at 8.5 Mbps. A separate subjective user study with 15 participants and approximately 200 played sessions further shows large MOS gains of up to 38% at 8.5 Mbps, and at 30 Mbps, our system maintains similar perceived quality while reducing bandwidth by about 50%, with MOS improving by 3.4% (OW2) and 4.6% (CS2) relative to baseline WebRTC. All these measurements include rendering, capture, encoding, network transmission, decoding, and GameSR upscaling; this indicates that the complete pipeline with GameSR integrated can satisfy real-time cloud gaming constraints in practice.
> > >
> > >   * **Dataset Generalization & Scalability:** We appreciate the concern about game and genre diversity and agree that this is an important dimension. In this work, we chose to focus on three shooter titles and to train GameSR per game, similar to how DLSS and FSR are typically tuned per title in practice. We also understand that CS2, OW2, and TF2 may feel similar, since they are all shooters. However, from a rendering and content perspective they are quite different: CS2 is a realistic tactical FPS on the Source 2 engine with high-contrast lighting and fine geometric detail; Overwatch 2 is a fast-paced hero shooter on Blizzard’s Overwatch engine with highly stylized characters and dense particle effects; and TF2 is an older class-based shooter on the original Source engine with flatter shading and heavy HUD elements. In the revision, we also add a fourth title, FIFA, a third-person sports game with long-range camera views, large uniform textures (pitch/grass), and different motion patterns (passes, shots, camera pans), and report its results in Sec .\~4.2 (Table\~2). As shown there, FIFA follows the same trend as our other games in terms of quality and efficiency, illustrating that GameSR extends naturally beyond FPS titles and that adding a new game requires only training on that game’s footage. In addition, these four titles span engines with very different support for industrial upscalers: recent engines where DLSS/FSR-style plugins are available, engines that only support FSR-like spatial upscalers because the forward-rendering data required by DLSS-style methods is not exposed, and a legacy engine (TF2) where neither DLSS nor modern FSR variants can be deployed. One might argue that heuristic upsamplers such as bicubic, bilinear, or Lanczos are sufficient for such older games, but our results in Table\~2 show that even for TF2, the bicubic baseline is consistently weaker than GameSR across all scales, both in reconstruction quality and perceptual metrics.
> > >
> > >   * **Inclusion of 2K & 4K results:** We focus on 1080p because, according to the Steam Hardware & Software Survey (October 2025), it is still the dominant gaming resolution (≈53% of players), while 1440p and 4K account for ≈20% and ≈5%, respectively, so 1080p remains the most practical operating point for real-time cloud gaming and streaming. To directly address the concern about higher resolutions, we have added 2K and 4K experiments in Sec. 4.2 (“Scalability and Generality to High-Resolution Upscaling,” Table 3): GameSR reaches up to 39.25 dB PSNR, SSIM \> 0.999, and VMAF \> 93 for 1080p→4K while keeping inference \< 16 ms per frame (real-time 60 FPS), and only 7 ms per frame for 720p→2K (≈143 FPS, suitable for 120 Hz). These results are obtained on unseen 2K/4K clips collected after training, showing that a model trained solely at 1080p **generalizes and scales** to substantially higher resolutions without fine-tuning, while maintaining high perceptual quality and low latency for practical high-resolution cloud gaming scenarios.

---

> > > > ### Author Response · Authors · 2025-11-19
> > > >
> > > > * **Evaluation Metric:** We apologize for the lack of clarity and will make the metric pipeline explicit. For all objective metrics (PSNR, SSIM, VMAF, LPIPS), the **high-resolution reference** is always the native game capture at full resolution (1080p in the main experiments, and 2K/4K in Sec. 4.2). The **low-resolution inputs** to GameSR are generated by downscaling these native captures: for 2x experiments we use 540p→1080p, for 3x we use 360p→1080p (and similarly 720p→2K, 1080p→4K, 720p→4K in the scalability experiments); in the cloud-gaming RD experiments, these LR sequences are additionally encoded and decoded to mimic a real streaming pipeline before being fed to GameSR. We then compute frame-wise PSNR/SSIM between the GameSR output and the native HR reference and report averages over each clip and game. VMAF is computed using the official Netflix implementation on the HR reference and the upscaled sequence, and LPIPS is computed using the standard PyTorch implementation; both have been widely shown to correlate well with human perception. Finally, our user study with 15 participants (Table 4\) yields high MOS scores (≈4.7/5) and consistent ratings across gamer/non-gamer groups, which supports that our VMAF/LPIPS trends indeed reflect perceived visual quality.
> > > >
> > > > * **Hardware Scalability:** We apologize for the confusion and will clarify the intended hardware targets in the revision. GameSR is designed as a client-side upscaler for commodity cloud-gaming devices ranging from smartphones to desktop GPUs, not only desktops. To make this explicit, we have added Sec. 4.2 (“Inference Evaluation Across Heterogeneous Hardware,” Fig. 6(c), where we benchmark both the original GameSR and a mobile-oriented variant, GameSR-M, across devices from an Adreno 750–class Samsung S24 mobile GPU (4.7 TFLOPS) up to RTX 40-series desktop GPUs. On the Adreno 750, GameSR-M already achieves real-time performance with 26.56 ms (X2) and 14.55 ms (X3) per 1080p frame, demonstrating feasibility on modern handhelds where NPUs/GPUs are typically underutilized during cloud gaming. On discrete GPUs, latency scales roughly inversely with compute: mid-range cards (RTX 4060 Ti / 4070 Ti / A4000) sustain 1.7–4.0 ms per frame, while high-end RTX 3090/4090 reach 2.3 ms and 1.3 ms, respectively. These results show that our design is explicitly hardware-scalable and can even run on resource-constrained devices.
> > > >
> > > > * **Clarifications:**
> > > >   * **GPU utilization:** Our intention was not to claim that lower GPU utilization is inherently “better optimized,” but to illustrate that rendering at 540p instead of 1080p naturally reduces the per-frame shading workload (about 0.52M vs. 2.07M pixels, a 4x reduction), after which GameSR recovers 1080p-quality imagery from this cheaper operating point.
> > > >
> > > >   * **Ablation study:** We already provide an ablation in Sec. 4.2 and App. A.6 (Table 7), where we remove ConvLSTM, PixelUnshuffle, and reparameterization and jointly measure PSNR/SSIM/LPIPS, latency, parameters, and memory. For example, removing ConvLSTM reduces latency to 3.05 ms and parameters to 65K but degrades PSNR by about 5 dB (37.99 → 32.99), and the ConvLSTM hidden state is kept locally on the client and never transmitted, so it adds no bandwidth overhead. We will surface these key numbers more prominently in the main paper.
> > > >
> > > >   * **Key/delta frames:** In our cloud-gaming setting, the client receives a live stream of decoded RGB frames, so GameSR is designed to operate per frame and is agnostic to codec internals such as I/P/B frame types. Given its low latency (approximately 3–6 ms per 1080p frame on mid-range GPUs), there is no need to upscale only keyframes and interpolate the rest, a strategy that is more suitable for offline VOD pipelines than for interactive, high-motion gaming.

---

> > > > > ### Comment · Reviewer_CgRw · 2025-11-25
> > > > >
> > > > > Thank you for your detailed response. I appreciate the explanations and additional paragraphs and results in the revision. I'd like to leave my comments below for a further discussion:
> > > > > 1. Thank you for your clarification of the positioning of the paper. Indeed, the paper provides an engine-agnostic SR framework, which is a different setting from existing engine-related SR frameworks requiring G-buffers. I acknowledge that this is more friendly for legacy games and cloud computing, although this may be less desired for modern AAA games that already has a mature engine integration pipeline.
> > > > > 2. The cloud gaming system looks very promising. In my opinion, the bandwidth reduction for cloud gaming is one of the biggest advantage of the proposed GameSR framework, and the provided cloud gaming system strengthens this aspect. I encourage the authors to provide more details of the cloud gaming system in the Appendix.
> > > > > 3. GPU utilization: If I'm understanding correctly, is the low GPU utilization because of the different purpose of the SR pipelines: DLSS/FSR strives to maximally boost the frame rate, while GameSR seeks to reduce computation with a reasonable frame rate? Is the GPU utilization comparison in Fig. 4(c) a same-FPS comparison? If GameSR can achieve the same FPS with lower GPU utilization, I'll agree that low GPU utilization is an advantage. If not. I'd still like to see the FPS number simultaneously together with the GPU utilization rate.

---

> > > > > > ### Author Response · Authors · 2025-12-02
> > > > > >
> > > > > > We appreciate the reviewer’s thoughtful follow-up on the cloud-gaming setup and GPU utilization, and we address these points below.
> > > > > >
> > > > > > - **On positioning and cloud gaming system details (Appendix. A8).**
> > > > > >
> > > > > >   - We are glad that the engine-agnostic setting and the cloud-gaming angle are clearer. Following your suggestion,  we have expanded the Appendix (A8) with more details of the aiortc/WebRTC system.
> > > > > >
> > > > > > - **On GPU utilization and FPS.**
> > > > > >
> > > > > >   - We thank the reviewer for this clarification request. In the configuration used for Fig. 4(c), the measured frame rates were approximately 123 FPS (native 1080p), 140 FPS (DLSS), 145 FPS (FSR), and about 125 FPS for GameSR. Thus, GameSR achieves essentially the same FPS as native while using only ≈82% GPU utilization, whereas native/DLSS/FSR run at similar or higher FPS but keep the GPU near 99–100% utilization. Both DLSS and FSR internally render at reduced resolution and then upscale to display resolution, and in typical usage the recovered shading budget is spent on higher FPS rather than reducing GPU load; this is consistent with our original text that “*DLSS/FSR are intended to boost framerate rather than reduce computation.*” Our point in Fig. 4(c) is therefore not that DLSS/FSR are inefficient or that high GPU utilization is inherently negative, but that the GameSR configuration can match native frame rate while leaving more GPU headroom. In a cloud-gaming deployment, where rendering is done on the server and upscaling is done on the client, this headroom can be converted into lower server-side rendering cost and reduced streaming bitrate by rendering at lower resolution and adding only a small SR overhead on the client. We have updated the main text around Fig. 4(c) to explicitly report these FPS numbers together with the utilization trends.

---

### Official Review · Reviewer_STmx · 2025-10-31

**Soundness:** 2
**Presentation:** 3
**Contribution:** 2
**Rating:** 4
**Confidence:** 4

**Summary:**

This paper presents a real-time edge-side super-resolution solution applicable to game scenarios. Through some improvements in network structures, it achieves performance similar to that of DLSS and FSR without accessing rendering engine data structures or modifying game source code.

**Strengths:**

1.The motivation of this article is commendable, as it attempts to apply deep learning super-resolution technology to the edge side and achieve performance similar to that of DLSS and FSR.
2.Write clearly and understandably.

**Weaknesses:**

As a method attempting to achieve real-time super-resolution on the edge side, merely comparing it with the existing classical image super-resolution methods is insufficient. There is no comparison with some existing edge-side super-resolution methods [1][2] and some lightweight video super-resolution methods [3][4][5][6].

There is a lack of necessary analysis and experiments regarding the ablation of the network structure design. For instance, what advantages does the ConvLSTM have compared to the residual block-based RNN structure [3]? Meanwhile, the report on the additional consumption required for transmitting the hidden state has not been provided yet.  Moreover, it is unclear to what extent the FEB module contributes to the super-resolution performance.
[1] edge–SR: Super–Resolution For The Masses. WACV 2022
[2] Super-Resolution by Predicting Offsets:An Ultra-Efficient Super-Resolution Network for Rasterized Images. ECCV 2022
[3] Efficient Video Super-Resolution through Recurrent Latent Space Propagation. ICCVW 19
[4] Stable Long-Term Recurrent Video Super-Resolution. CVPR 22
[5] Structured Sparsity Learning for Efficient Video Super-Resolution. CVPR 23
[6] Efficient Video Super-Resolution for Real-time Rendering with Decoupled G-buffer Guidance. CVPR 25

**Questions:**

Please refer to the weakness part.

---

> ### Author Response · Authors · 2025-11-19
>
> We appreciate the reviewers’ constructive feedback. We have addressed the main concerns as follows:
>
> 1. **End-to-end cloud gaming system (Sec. 4.3).**
>     We now briefly describe our complete WebRTC-based system (RL-based rate controller \+ GameSR after H.264 decode). We also summarize the end-to-end results from over 50 CS2/OW2 gaming sessions under realistic latency and bandwidth conditions, including bandwidth/CPU/GPU savings and MOS gains.
>
> 2. **Expanded evaluation and scalability (Secs. 4.2, Table 3, Fig. 6(c)).**
>    We add 2K/4K experiments, heterogeneous hardware results from mobile GPUs (Adreno 750 / Samsung S24) to RTX GPUs, and a new game (FIFA24), showing that GameSR generalizes beyond first-person shooter games, scales to high resolutions, and supports various devices.
>
> 3. **Positioning and engine-agnostic design (Secs. 3.1).**
>    We clarify that GameSR is an engine- and codec-agnostic post-render, post-decoder upscaler for both local rendering and cloud gaming, complementing DLSS/FSR by working on legacy, closed-source, and multi-engine titles where render buffers are unavailable.
>
> 4. **Relation to prior edge / video SR methods (Sec. 2).**
>     We extend related work to discuss edge-SR, engine-integrated SR (SRPO, RDG, FuseSR, MNSS, Deep Fourier), and recurrent VSR (RLSP, MRVSR, SSL), explaining why their engine coupling, fixed 4X scale, and runtime make them complementary rather than direct baselines for our 2X/3X, 1080p, post-decoder setting.
>
> 5. **Metrics, perceptual quality, and ablations (Secs. 4.1, 4.2, App. A.6).**
>     We clarify how PSNR/SSIM/VMAF/LPIPS are computed, connect them to our 15-participant user study, and highlight existing ablations of ConvLSTM, PixelUnshuffle, and reparameterization, as well as our updated discussion of GPU utilization and frame-wise operation.
>
> More details about the revisions are provided below:
>
> * **Positioning of GameSR:** We clarify the positioning of GameSR in the revision. Our goal is to support **both local rendering and cloud gaming**, but always in an **engine-agnostic** way: GameSR runs purely on RGB frames and does not require access to G-buffers, motion vectors, depth, or vendor-specific hardware features (e.g., DLSS on RTX with dedicated integration). This makes it applicable to a much broader set of titles than current industrial upscalers: it can be used with legacy games such as TF2 on the original Source engine, with engines that do not expose the rendering buffers or motion/depth signals that modern DLSS/FSR plugins assume, and with cloud-gaming pipelines where only decoded video is available and transmitting extra render data would be too costly. Our **main comparison focus** is therefore on widely deployed game upsamplers (DLSS/FSR) because they represent the dominant practical baselines for high-quality gaming SR today; GameSR is designed to *complement* these methods by offering a vendor- and engine-independent alternative that works on both local PCs/consoles and cloud clients, including for titles and platforms where engine-integrated solutions are not feasible.
>
>
> * **On the practicality of GameSR and the advantage of an engine-free design:** We agree that engine-level signals (G-buffers, motion, depth) are very powerful and that for AAA titles with existing DLSS/FSR-style integrations the quality–integration trade-off is often acceptable. GameSR, however, is explicitly targeted at cloud gaming and legacy or multi-engine catalogues, where clients only see encoded/decoded RGB frames and engine buffers are either inaccessible (closed-source, older engines) or too costly to transmit at scale. In these settings, just integrating it into the engine is not realistic, so an engine-agnostic post-decode upscaler is required.

---

> > ### Author Response · Authors · 2025-11-19
> >
> > * **Comparisons against other works mentioned by the reviewers:**
> >
> >   * Our setting is fundamentally different from **GameStreamSR**, so we do not view it as a directly comparable baseline. GameStreamSR is an engine-dependent, depth-guided RoI upscaler: it applies a heavy SR network (instantiated with existing models such as EDSR) only to a small depth-defined region, and upscales the rest of the frame with bilinear interpolation. This design is tailored to mobile cloud gaming with access to render buffers and control over the engine, but leaves large portions of the image at bilinear quality (including HUD, distant targets, and text) and cannot be used in generic cloud-gaming clients that only see encoded/decoded RGB frames. In contrast, GameSR is a lightweight, full-frame post-render, post-decoder upscaler that operates purely on RGB video and can be deployed on both local and cloud clients, including legacy and closed-source titles where engine data are unavailable.
> >
> >   * **edge-SR** is built around very shallow (often 1–3 layer) single-image models, designed as drop-in replacements for classic upscalers on generic natural images. Modern 3D game frames are far more complex (high motion, fine geometry, HUDs, compression artifacts), and even **deeper** state-of-the-art SR baselines in our experiments struggle to fully recover accurate details. Given this, we do not expect a 1–3 layer image-only architecture to be competitive in our high-resolution, temporally demanding games.
> >
> >   * **SRPO** (ECCV’22), **RDG** (CVPR’25), **FuseSR**, **MNSS**, and **the Deep Fourier-based arbitrary-scale SR** method all target a different operating point than GameSR: they are tightly integrated into the real-time renderer and rely on internal engine signals such as high-resolution G-buffers, motion vectors, depth, and other per-pixel attributes. This requires modifying or instrumenting the game engine and is mainly practical when one controls the renderer (e.g., in-house AAA titles). By contrast, GameSR is deliberately designed as a **post-render, post-decoder** upscaler that operates solely on RGB frames: it can run (i) after video decode in a cloud-gaming client, and (ii) after a lower-resolution render target is resolved locally (e.g., 540p/720p→1080p) without any access to engine buffers or vendor-specific hardware. This makes it applicable to legacy and closed-source games, engines without forward-rendering data exposed, and generic streaming platforms where only decoded video is available. Moreover, these engine-coupled methods are typically evaluated in controlled rendering demos or author-created scenes rather than on long, recorded gameplay streams under encoding and network constraints as in our experiments. For these reasons, we view them as complementary engine-integrated approaches and do not treat them as directly comparable baselines for the post-render/post-decoder deployment scenario that GameSR targets.
> >
> >   * Regarding *Efficient Video Super-Resolution through Recurrent Latent Space Propagation* (**RLSP**), *Stable Long-Term Recurrent Video Super-Resolution* (**MRVSR**), and ***Structured Sparsity Learning for Efficient Video Super-Resolution***, we agree that they are important recurrent VSR baselines, but they target a very different domain than ours. All three are fixed x4 models evaluated on small LR inputs (typically 180×320→720p), whereas in gaming and modern upscalers such as DLSS/FSR the practical setting for 1080p is closer to 2x (e.g., 960×540→1080p or 720p→1080p); pushing to 4x from very low LR (e.g., 270p→1080p) is generally avoided because quality and object visibility degrade noticeably in games. A single fixed 4x scale also offers limited flexibility for any intelligent rate controller that needs to adapt rendering resolution to network congestion (e.g., switching between 360p/540p/720p with corresponding 2x/3x upscaling factors), which a 4x-only design cannot support. In addition, the reported runtimes for these models are on the order of tens of milliseconds per frame at 180×320 on high-end GPUs (TITAN RTX, TITAN Xp, V100), whereas GameSR achieves 2x/3x upscaling for full-HD gaming streams (540p/720p→1080p) in about 4–5 ms per frame on a similar GPU in terms of power with only 138K parameters. Adapting these architectures to our 1080p, 2x/3x, 60 fps setting would require substantial redesign and still provide less control to an adaptive rate controller, so we treat them as complementary general VSR work rather than directly comparable baselines for the deployment we target.

---

> > > ### Author Response · Authors · 2025-11-19
> > >
> > > * **Expanded Evaluation:**
> > >
> > >   * **E2E Cloud Gaming System:** Our original submission focused on the SR component in isolation, but we have, in fact, implemented and evaluated a complete end-to-end cloud gaming system built on aiortc/WebRTC. The system includes an RL-based Joint Optimizer with a Complexity Analyzer on the server that selects the rendering resolution and encoder rate once per group of pictures (GOP), and GameSR running on the client-side after H.264 decode. In the revised paper, we add Sec. 4.3, “End-to-End Cloud Gaming System”, where we summarize this design and report results over 50 gaming sessions of CS2 and OW2 under realistic latency traces and two bandwidth regimes (30 Mbps and 8.5 Mbps). Compared to baseline WebRTC, our framework achieves savings of up to 50%, 62%, and 41% in bandwidth, CPU, and GPU requirements, respectively, while maintaining VMAF close to 90 and significantly improving quality at 8.5 Mbps. A separate subjective user study with 15 participants and approximately 200 played sessions further shows large MOS gains of up to 38% at 8.5 Mbps, and at 30 Mbps, our system maintains similar perceived quality while reducing bandwidth by about 50%, with MOS improving by 3.4% (OW2) and 4.6% (CS2) relative to baseline WebRTC. All these measurements include rendering, capture, encoding, network transmission, decoding, and GameSR upscaling; this indicates that the complete pipeline with GameSR integrated can satisfy real-time cloud gaming constraints in practice.
> > >
> > >   * **Dataset Generalization & Scalability:** We appreciate the concern about game and genre diversity and agree that this is an important dimension. In this work, we chose to focus on three shooter titles and to train GameSR per game, similar to how DLSS and FSR are typically tuned per title in practice. We also understand that CS2, OW2, and TF2 may feel similar, since they are all shooters. However, from a rendering and content perspective they are quite different: CS2 is a realistic tactical FPS on the Source 2 engine with high-contrast lighting and fine geometric detail; Overwatch 2 is a fast-paced hero shooter on Blizzard’s Overwatch engine with highly stylized characters and dense particle effects; and TF2 is an older class-based shooter on the original Source engine with flatter shading and heavy HUD elements. In the revision, we also add a fourth title, FIFA, a third-person sports game with long-range camera views, large uniform textures (pitch/grass), and different motion patterns (passes, shots, camera pans), and report its results in Sec .\~4.2 (Table\~2). As shown there, FIFA follows the same trend as our other games in terms of quality and efficiency, illustrating that GameSR extends naturally beyond FPS titles and that adding a new game requires only training on that game’s footage. In addition, these four titles span engines with very different support for industrial upscalers: recent engines where DLSS/FSR-style plugins are available, engines that only support FSR-like spatial upscalers because the forward-rendering data required by DLSS-style methods is not exposed, and a legacy engine (TF2) where neither DLSS nor modern FSR variants can be deployed. One might argue that heuristic upsamplers such as bicubic, bilinear, or Lanczos are sufficient for such older games, but our results in Table\~2 show that even for TF2, the bicubic baseline is consistently weaker than GameSR across all scales, both in reconstruction quality and perceptual metrics.
> > >
> > >   * **Inclusion of 2K & 4K results:** We focus on 1080p because, according to the Steam Hardware & Software Survey (October 2025), it is still the dominant gaming resolution (≈53% of players), while 1440p and 4K account for ≈20% and ≈5%, respectively, so 1080p remains the most practical operating point for real-time cloud gaming and streaming. To directly address the concern about higher resolutions, we have added 2K and 4K experiments in Sec. 4.2 (“Scalability and Generality to High-Resolution Upscaling,” Table 3): GameSR reaches up to 39.25 dB PSNR, SSIM \> 0.999, and VMAF \> 93 for 1080p→4K while keeping inference \< 16 ms per frame (real-time 60 FPS), and only 7 ms per frame for 720p→2K (≈143 FPS, suitable for 120 Hz). These results are obtained on unseen 2K/4K clips collected after training, showing that a model trained solely at 1080p **generalizes and scales** to substantially higher resolutions without fine-tuning, while maintaining high perceptual quality and low latency for practical high-resolution cloud gaming scenarios.

---

> ### Author Response · Authors · 2025-11-19
>
> * **Evaluation Metric:** We apologize for the lack of clarity and will make the metric pipeline explicit. For all objective metrics (PSNR, SSIM, VMAF, LPIPS), the **high-resolution reference** is always the native game capture at full resolution (1080p in the main experiments, and 2K/4K in Sec. 4.2). The **low-resolution inputs** to GameSR are generated by downscaling these native captures: for 2x experiments we use 540p→1080p, for 3x we use 360p→1080p (and similarly 720p→2K, 1080p→4K, 720p→4K in the scalability experiments); in the cloud-gaming RD experiments, these LR sequences are additionally encoded and decoded to mimic a real streaming pipeline before being fed to GameSR. We then compute frame-wise PSNR/SSIM between the GameSR output and the native HR reference and report averages over each clip and game. VMAF is computed using the official Netflix implementation on the HR reference and the upscaled sequence, and LPIPS is computed using the standard PyTorch implementation; both have been widely shown to correlate well with human perception. Finally, our user study with 15 participants (Table 4\) yields high MOS scores (≈4.7/5) and consistent ratings across gamer/non-gamer groups, which supports that our VMAF/LPIPS trends indeed reflect perceived visual quality.
>
> * **Hardware Scalability:** We apologize for the confusion and will clarify the intended hardware targets in the revision. GameSR is designed as a client-side upscaler for commodity cloud-gaming devices ranging from smartphones to desktop GPUs, not only desktops. To make this explicit, we have added Sec. 4.2 (“Inference Evaluation Across Heterogeneous Hardware,” Fig. 6(c), where we benchmark both the original GameSR and a mobile-oriented variant, GameSR-M, across devices from an Adreno 750–class Samsung S24 mobile GPU (4.7 TFLOPS) up to RTX 40-series desktop GPUs. On the Adreno 750, GameSR-M already achieves real-time performance with 26.56 ms (X2) and 14.55 ms (X3) per 1080p frame, demonstrating feasibility on modern handhelds where NPUs/GPUs are typically underutilized during cloud gaming. On discrete GPUs, latency scales roughly inversely with compute: mid-range cards (RTX 4060 Ti / 4070 Ti / A4000) sustain 1.7–4.0 ms per frame, while high-end RTX 3090/4090 reach 2.3 ms and 1.3 ms, respectively. These results show that our design is explicitly hardware-scalable and can even run on resource-constrained devices.
>
> * **Clarifications:**
>   * **GPU utilization:** In the configuration used for Fig. 4(c), the measured frame rates were approximately 123 FPS (native 1080p), 140 FPS (DLSS), 145 FPS (FSR), and about 125 FPS for GameSR. Thus, GameSR achieves essentially the same FPS as native while using only ≈82% GPU utilization, whereas native/DLSS/FSR run at similar or higher FPS but keep the GPU near 99–100% utilization. Both DLSS and FSR internally render at reduced resolution and then upscale to display resolution, and in typical usage the recovered shading budget is spent on higher FPS rather than reducing GPU load; this is consistent with our original text that “*DLSS/FSR are intended to boost framerate rather than reduce computation.*” Our point in Fig. 4(c) is therefore not that DLSS/FSR are inefficient or that high GPU utilization is inherently negative, but that the GameSR configuration can match native frame rate while leaving more GPU headroom. In a cloud-gaming deployment, where rendering is done on the server and upscaling is done on the client, this headroom can be converted into lower server-side rendering cost and reduced streaming bitrate by rendering at lower resolution and adding only a small SR overhead on the client. We have updated the main text around Fig. 4(c) to explicitly report these FPS numbers together with the utilization trends.
>
>   * **Ablation study:** We already provide an ablation in Sec. 4.2 and App. A.6 (Table 7), where we remove ConvLSTM, PixelUnshuffle, and reparameterization and jointly measure PSNR/SSIM/LPIPS, latency, parameters, and memory. For example, removing ConvLSTM reduces latency to 3.05 ms and parameters to 65K but degrades PSNR by about 5 dB (37.99 → 32.99), and the ConvLSTM hidden state is kept locally on the client and never transmitted, so it adds no bandwidth overhead. We will surface these key numbers more prominently in the main paper.
>
>   * **Key/delta frames:** In our cloud-gaming setting, the client receives a live stream of decoded RGB frames, so GameSR is designed to operate per frame and is agnostic to codec internals such as I/P/B frame types. Given its low latency (approximately 3–6 ms per 1080p frame on mid-range GPUs), there is no need to upscale only keyframes and interpolate the rest, a strategy that is more suitable for offline VOD pipelines than for interactive, high-motion gaming.

---

### Official Review · Reviewer_GWsu · 2025-11-01

**Soundness:** 2
**Presentation:** 3
**Contribution:** 2
**Rating:** 2
**Confidence:** 4

**Summary:**

The paper proposes GameSR, an efficient approach for super-scaling video frames in cloud gaming. Unlike industrial solutions that rely on accessing in-game data such as motion vectors and depth maps, GameSR works without requiring direct access to the game engine. The framework consists of three main components: feature extraction, feature learning, and temporal learning. According to the evaluation, GameSR achieves visual quality comparable to industrial solutions while being significantly more efficient than existing CNN-based methods.

**Strengths:**

+ The paper explores an interesting and relevant problem domain in cloud gaming.
+ The writing and organization are clear and easy to follow.
+ GameSR demonstrates on-par or even superior quality and efficiency compared to both state-of-the-art research baselines and industrial solutions.

**Weaknesses:**

- Target Resolution: My major concern is on the experimental resolution setup. The paper targets high-resolution cloud gaming but only evaluates upscaling to 1080p. For desktop-level hardware such as an NVIDIA RTX A4000, 1080p is relatively low, especially given that high-end gaming commonly targets 2K or 4K resolutions. Even for handheld devices, prior work ([GameStreamSR: Enabling Neural-Augmented Game Streaming on Commodity Mobile Platforms](https://ieeexplore.ieee.org/abstract/document/10609685)) has already moved toward higher target resolutions (e.g., 1440p). Therefore, the chosen setting may not be representative and reasonable for real-world use cases, and the reported quality, frame rate, and memory usage might not be valid for the intended scenario.

- Dataset Generalizability: The effectiveness of the method may heavily depend on the type of game and the way it is played. The authors seem to use a custom-collected dataset rather than a public benchmark. It is unclear if the captured gameplay scenarios are diverse enough or if they might unintentionally favor the proposed method.

- Scope of Evaluation: The paper only evaluates the super-resolution component itself. However, for cloud gaming, the end-to-end latency—including rendering, encoding, network transmission, decoding, and upscaling—is the key metric for real-time performance. Without this measurement, it is difficult to assess the practical benefit of GameSR.

- Limited Game Selection: GameSR is evaluated on only three games. This may be insufficient to demonstrate its robustness and general-purpose applicability across different game genres, art styles, and game engines.

**Questions:**

1. Could you please clarify the methodology for measuring the quality metrics (PSNR, SSIM, VMAF and LPIPS)? Specifically, what frames were used as the high-resolution ground truth reference, and what were the corresponding low-resolution inputs provided to GameSR?
2. The paper appears to target desktop-level hardware for super resolution inference, but the experimental setup could be stated more clearly. Could you clarify the intended hardware target(s)? How would you expect GameSR to perform on more resource-constrained handheld devices?
3. While the super-resolution performance result is detailed, what is the expected breakdown of latencies in a real-world, end-to-end deployment (i.e., including rendering, encoding, transmission, decoding, etc.)? Can the entire pipeline, with GameSR integrated, realistically achieve the real-time performance required for interactive high resolution cloud gaming?
4. How does GameSR's performance (in terms of both quality and processing speed) scale when targeting higher resolutions, such as 2K or 4K, which are more realistic for the described desktop gaming scenario?

---

> ### Author Response · Authors · 2025-11-19
>
> #### **Reviewer Reply:**
>
> We appreciate the reviewers’ constructive feedback. We have addressed the main concerns as follows:
>
> 1. **End-to-end cloud gaming system (Sec. 4.3).**
>     We now briefly describe our complete WebRTC-based system (RL-based rate controller \+ GameSR after H.264 decode). We also summarize the end-to-end results from over 50 CS2/OW2 gaming sessions under realistic latency and bandwidth conditions, including bandwidth/CPU/GPU savings and MOS gains.
>
> 2. **Expanded evaluation and scalability (Secs. 4.2, Table 3, Fig. 6(c)).**
>    We add 2K/4K experiments, heterogeneous hardware results from mobile GPUs (Adreno 750 / Samsung S24) to RTX GPUs, and a new game (FIFA24), showing that GameSR generalizes beyond first-person shooter games, scales to high resolutions, and supports various devices.
>
> 3. **Positioning and engine-agnostic design (Secs. 3.1).**
>    We clarify that GameSR is an engine- and codec-agnostic post-render, post-decoder upscaler for both local rendering and cloud gaming, complementing DLSS/FSR by working on legacy, closed-source, and multi-engine titles where render buffers are unavailable.
>
> 4. **Relation to prior edge / video SR methods (Sec. 2).**
>     We extend related work to discuss edge-SR, engine-integrated SR (SRPO, RDG, FuseSR, MNSS, Deep Fourier), and recurrent VSR (RLSP, MRVSR, SSL), explaining why their engine coupling, fixed 4X scale, and runtime make them complementary rather than direct baselines for our 2X/3X, 1080p, post-decoder setting.
>
> 5. **Metrics, perceptual quality, and ablations (Secs. 4.1, 4.2, App. A.6).**
>     We clarify how PSNR/SSIM/VMAF/LPIPS are computed, connect them to our 15-participant user study, and highlight existing ablations of ConvLSTM, PixelUnshuffle, and reparameterization, as well as our updated discussion of GPU utilization and frame-wise operation.
>
> More details about the revisions are provided below:
>
> * **Positioning of GameSR:** We clarify the positioning of GameSR in the revision. Our goal is to support **both local rendering and cloud gaming**, but always in an **engine-agnostic** way: GameSR runs purely on RGB frames and does not require access to G-buffers, motion vectors, depth, or vendor-specific hardware features (e.g., DLSS on RTX with dedicated integration). This makes it applicable to a much broader set of titles than current industrial upscalers: it can be used with legacy games such as TF2 on the original Source engine, with engines that do not expose the rendering buffers or motion/depth signals that modern DLSS/FSR plugins assume, and with cloud-gaming pipelines where only decoded video is available and transmitting extra render data would be too costly. Our **main comparison focus** is therefore on widely deployed game upsamplers (DLSS/FSR) because they represent the dominant practical baselines for high-quality gaming SR today; GameSR is designed to *complement* these methods by offering a vendor- and engine-independent alternative that works on both local PCs/consoles and cloud clients, including for titles and platforms where engine-integrated solutions are not feasible.
>
>
> * **On the practicality of GameSR and the advantage of an engine-free design:** We agree that engine-level signals (G-buffers, motion, depth) are very powerful and that for AAA titles with existing DLSS/FSR-style integrations the quality–integration trade-off is often acceptable. GameSR, however, is explicitly targeted at cloud gaming and legacy or multi-engine catalogues, where clients only see encoded/decoded RGB frames and engine buffers are either inaccessible (closed-source, older engines) or too costly to transmit at scale. In these settings, just integrating it into the engine is not realistic, so an engine-agnostic post-decode upscaler is required.

---

> ### Author Response · Authors · 2025-11-19
>
> * **Comparisons against other works mentioned by the reviewers:**
>
>   * Our setting is fundamentally different from **GameStreamSR**, so we do not view it as a directly comparable baseline. GameStreamSR is an engine-dependent, depth-guided RoI upscaler: it applies a heavy SR network (instantiated with existing models such as EDSR) only to a small depth-defined region, and upscales the rest of the frame with bilinear interpolation. This design is tailored to mobile cloud gaming with access to render buffers and control over the engine, but leaves large portions of the image at bilinear quality (including HUD, distant targets, and text) and cannot be used in generic cloud-gaming clients that only see encoded/decoded RGB frames. In contrast, GameSR is a lightweight, full-frame post-render, post-decoder upscaler that operates purely on RGB video and can be deployed on both local and cloud clients, including legacy and closed-source titles where engine data are unavailable.
>
>   * **edge-SR** is built around very shallow (often 1–3 layer) single-image models, designed as drop-in replacements for classic upscalers on generic natural images. Modern 3D game frames are far more complex (high motion, fine geometry, HUDs, compression artifacts), and even **deeper** state-of-the-art SR baselines in our experiments struggle to fully recover accurate details. Given this, we do not expect a 1–3 layer image-only architecture to be competitive in our high-resolution, temporally demanding games.
>
>   * **SRPO** (ECCV’22), **RDG** (CVPR’25), **FuseSR**, **MNSS**, and **the Deep Fourier-based arbitrary-scale SR** method all target a different operating point than GameSR: they are tightly integrated into the real-time renderer and rely on internal engine signals such as high-resolution G-buffers, motion vectors, depth, and other per-pixel attributes. This requires modifying or instrumenting the game engine and is mainly practical when one controls the renderer (e.g., in-house AAA titles). By contrast, GameSR is deliberately designed as a **post-render, post-decoder** upscaler that operates solely on RGB frames: it can run (i) after video decode in a cloud-gaming client, and (ii) after a lower-resolution render target is resolved locally (e.g., 540p/720p→1080p) without any access to engine buffers or vendor-specific hardware. This makes it applicable to legacy and closed-source games, engines without forward-rendering data exposed, and generic streaming platforms where only decoded video is available. Moreover, these engine-coupled methods are typically evaluated in controlled rendering demos or author-created scenes rather than on long, recorded gameplay streams under encoding and network constraints as in our experiments. For these reasons, we view them as complementary engine-integrated approaches and do not treat them as directly comparable baselines for the post-render/post-decoder deployment scenario that GameSR targets.
>
>   * Regarding *Efficient Video Super-Resolution through Recurrent Latent Space Propagation* (**RLSP**), *Stable Long-Term Recurrent Video Super-Resolution* (**MRVSR**), and ***Structured Sparsity Learning for Efficient Video Super-Resolution***, we agree that they are important recurrent VSR baselines, but they target a very different domain than ours. All three are fixed x4 models evaluated on small LR inputs (typically 180×320→720p), whereas in gaming and modern upscalers such as DLSS/FSR the practical setting for 1080p is closer to 2x (e.g., 960×540→1080p or 720p→1080p); pushing to 4x from very low LR (e.g., 270p→1080p) is generally avoided because quality and object visibility degrade noticeably in games. A single fixed 4x scale also offers limited flexibility for any intelligent rate controller that needs to adapt rendering resolution to network congestion (e.g., switching between 360p/540p/720p with corresponding 2x/3x upscaling factors), which a 4x-only design cannot support. In addition, the reported runtimes for these models are on the order of tens of milliseconds per frame at 180×320 on high-end GPUs (TITAN RTX, TITAN Xp, V100), whereas GameSR achieves 2x/3x upscaling for full-HD gaming streams (540p/720p→1080p) in about 4–5 ms per frame on a similar GPU in terms of power with only 138K parameters. Adapting these architectures to our 1080p, 2x/3x, 60 fps setting would require substantial redesign and still provide less control to an adaptive rate controller, so we treat them as complementary general VSR work rather than directly comparable baselines for the deployment we target.

---

> ### Author Response · Authors · 2025-11-19
>
> * **Expanded Evaluation:**
>
>   * **E2E Cloud Gaming System:** Our original submission focused on the SR component in isolation, but we have, in fact, implemented and evaluated a complete end-to-end cloud gaming system built on aiortc/WebRTC. The system includes an RL-based Joint Optimizer with a Complexity Analyzer on the server that selects the rendering resolution and encoder rate once per group of pictures (GOP), and GameSR running on the client-side after H.264 decode. In the revised paper, we add Sec. 4.3, “End-to-End Cloud Gaming System”, where we summarize this design and report results over 50 gaming sessions of CS2 and OW2 under realistic latency traces and two bandwidth regimes (30 Mbps and 8.5 Mbps). Compared to baseline WebRTC, our framework achieves savings of up to 50%, 62%, and 41% in bandwidth, CPU, and GPU requirements, respectively, while maintaining VMAF close to 90 and significantly improving quality at 8.5 Mbps. A separate subjective user study with 15 participants and approximately 200 played sessions further shows large MOS gains of up to 38% at 8.5 Mbps, and at 30 Mbps, our system maintains similar perceived quality while reducing bandwidth by about 50%, with MOS improving by 3.4% (OW2) and 4.6% (CS2) relative to baseline WebRTC. All these measurements include rendering, capture, encoding, network transmission, decoding, and GameSR upscaling; this indicates that the complete pipeline with GameSR integrated can satisfy real-time cloud gaming constraints in practice.
>
>   * **Dataset Generalization & Scalability:** We appreciate the concern about game and genre diversity and agree that this is an important dimension. In this work, we chose to focus on three shooter titles and to train GameSR per game, similar to how DLSS and FSR are typically tuned per title in practice. We also understand that CS2, OW2, and TF2 may feel similar, since they are all shooters. However, from a rendering and content perspective they are quite different: CS2 is a realistic tactical FPS on the Source 2 engine with high-contrast lighting and fine geometric detail; Overwatch 2 is a fast-paced hero shooter on Blizzard’s Overwatch engine with highly stylized characters and dense particle effects; and TF2 is an older class-based shooter on the original Source engine with flatter shading and heavy HUD elements. In the revision, we also add a fourth title, FIFA, a third-person sports game with long-range camera views, large uniform textures (pitch/grass), and different motion patterns (passes, shots, camera pans), and report its results in Sec .\~4.2 (Table\~2). As shown there, FIFA follows the same trend as our other games in terms of quality and efficiency, illustrating that GameSR extends naturally beyond FPS titles and that adding a new game requires only training on that game’s footage. In addition, these four titles span engines with very different support for industrial upscalers: recent engines where DLSS/FSR-style plugins are available, engines that only support FSR-like spatial upscalers because the forward-rendering data required by DLSS-style methods is not exposed, and a legacy engine (TF2) where neither DLSS nor modern FSR variants can be deployed. One might argue that heuristic upsamplers such as bicubic, bilinear, or Lanczos are sufficient for such older games, but our results in Table\~2 show that even for TF2, the bicubic baseline is consistently weaker than GameSR across all scales, both in reconstruction quality and perceptual metrics.
>
>   * **Inclusion of 2K & 4K results:** We focus on 1080p because, according to the Steam Hardware & Software Survey (October 2025), it is still the dominant gaming resolution (≈53% of players), while 1440p and 4K account for ≈20% and ≈5%, respectively, so 1080p remains the most practical operating point for real-time cloud gaming and streaming. To directly address the concern about higher resolutions, we have added 2K and 4K experiments in Sec. 4.2 (“Scalability and Generality to High-Resolution Upscaling,” Table 3): GameSR reaches up to 39.25 dB PSNR, SSIM \> 0.999, and VMAF \> 93 for 1080p→4K while keeping inference \< 16 ms per frame (real-time 60 FPS), and only 7 ms per frame for 720p→2K (≈143 FPS, suitable for 120 Hz). These results are obtained on unseen 2K/4K clips collected after training, showing that a model trained solely at 1080p **generalizes and scales** to substantially higher resolutions without fine-tuning, while maintaining high perceptual quality and low latency for practical high-resolution cloud gaming scenarios.

---

> ### Author Response · Authors · 2025-11-19
>
> * **Evaluation Metric:** We apologize for the lack of clarity and will make the metric pipeline explicit. For all objective metrics (PSNR, SSIM, VMAF, LPIPS), the **high-resolution reference** is always the native game capture at full resolution (1080p in the main experiments, and 2K/4K in Sec. 4.2). The **low-resolution inputs** to GameSR are generated by downscaling these native captures: for 2x experiments we use 540p→1080p, for 3x we use 360p→1080p (and similarly 720p→2K, 1080p→4K, 720p→4K in the scalability experiments); in the cloud-gaming RD experiments, these LR sequences are additionally encoded and decoded to mimic a real streaming pipeline before being fed to GameSR. We then compute frame-wise PSNR/SSIM between the GameSR output and the native HR reference and report averages over each clip and game. VMAF is computed using the official Netflix implementation on the HR reference and the upscaled sequence, and LPIPS is computed using the standard PyTorch implementation; both have been widely shown to correlate well with human perception. Finally, our user study with 15 participants (Table 4\) yields high MOS scores (≈4.7/5) and consistent ratings across gamer/non-gamer groups, which supports that our VMAF/LPIPS trends indeed reflect perceived visual quality.
>
> * **Hardware Scalability:** We apologize for the confusion and will clarify the intended hardware targets in the revision. GameSR is designed as a client-side upscaler for commodity cloud-gaming devices ranging from smartphones to desktop GPUs, not only desktops. To make this explicit, we have added Sec. 4.2 (“Inference Evaluation Across Heterogeneous Hardware,” Fig. 6(c), where we benchmark both the original GameSR and a mobile-oriented variant, GameSR-M, across devices from an Adreno 750–class Samsung S24 mobile GPU (4.7 TFLOPS) up to RTX 40-series desktop GPUs. On the Adreno 750, GameSR-M already achieves real-time performance with 26.56 ms (X2) and 14.55 ms (X3) per 1080p frame, demonstrating feasibility on modern handhelds where NPUs/GPUs are typically underutilized during cloud gaming. On discrete GPUs, latency scales roughly inversely with compute: mid-range cards (RTX 4060 Ti / 4070 Ti / A4000) sustain 1.7–4.0 ms per frame, while high-end RTX 3090/4090 reach 2.3 ms and 1.3 ms, respectively. These results show that our design is explicitly hardware-scalable and can even run on resource-constrained devices.
>
> * **Clarifications:**
>   * **GPU utilization:** In the configuration used for Fig. 4(c), the measured frame rates were approximately 123 FPS (native 1080p), 140 FPS (DLSS), 145 FPS (FSR), and about 125 FPS for GameSR. Thus, GameSR achieves essentially the same FPS as native while using only ≈82% GPU utilization, whereas native/DLSS/FSR run at similar or higher FPS but keep the GPU near 99–100% utilization. Both DLSS and FSR internally render at reduced resolution and then upscale to display resolution, and in typical usage the recovered shading budget is spent on higher FPS rather than reducing GPU load; this is consistent with our original text that “*DLSS/FSR are intended to boost framerate rather than reduce computation.*” Our point in Fig. 4(c) is therefore not that DLSS/FSR are inefficient or that high GPU utilization is inherently negative, but that the GameSR configuration can match native frame rate while leaving more GPU headroom. In a cloud-gaming deployment, where rendering is done on the server and upscaling is done on the client, this headroom can be converted into lower server-side rendering cost and reduced streaming bitrate by rendering at lower resolution and adding only a small SR overhead on the client. We have updated the main text around Fig. 4(c) to explicitly report these FPS numbers together with the utilization trends.
>
>   * **Ablation study:** We already provide an ablation in Sec. 4.2 and App. A.6 (Table 7), where we remove ConvLSTM, PixelUnshuffle, and reparameterization and jointly measure PSNR/SSIM/LPIPS, latency, parameters, and memory. For example, removing ConvLSTM reduces latency to 3.05 ms and parameters to 65K but degrades PSNR by about 5 dB (37.99 → 32.99), and the ConvLSTM hidden state is kept locally on the client and never transmitted, so it adds no bandwidth overhead. We will surface these key numbers more prominently in the main paper.
>
>   * **Key/delta frames:** In our cloud-gaming setting, the client receives a live stream of decoded RGB frames, so GameSR is designed to operate per frame and is agnostic to codec internals such as I/P/B frame types. Given its low latency (approximately 3–6 ms per 1080p frame on mid-range GPUs), there is no need to upscale only keyframes and interpolate the rest, a strategy that is more suitable for offline VOD pipelines than for interactive, high-motion gaming.

---

### Author Response · Authors · 2025-12-02

We appreciate the reviewers’ constructive feedback. For the new area chair, we summarize below how we have addressed the main concerns:

1. **End-to-end cloud gaming system (Sec. 4.3, App A.8.).**
    We now briefly describe our complete WebRTC-based system (RL-based rate controller \+ GameSR after H.264 decode). We also summarize the end-to-end results from over 50 CS2/OW2 gaming sessions under realistic latency and bandwidth conditions, including bandwidth/CPU/GPU savings and MOS gains.

2. **Expanded evaluation and scalability (Secs. 4.2, Table 3, Fig. 6(c)).**
   We add 2K/4K experiments, heterogeneous hardware results from mobile GPUs (Adreno 750 / Samsung S24) to RTX GPUs, and a new game (FIFA24), showing that GameSR generalizes beyond first-person shooter games, scales to high resolutions, and supports various devices.

3. **Positioning and engine-agnostic design (Secs. 3.1).**
   We clarify that GameSR is an engine- and codec-agnostic post-render, post-decoder upscaler for both local rendering and cloud gaming, complementing DLSS/FSR by working on legacy, closed-source, and multi-engine titles where render buffers are unavailable.

4. **Relation to prior edge / video SR methods (Sec. 2).**
    We extend related work to discuss edge-SR, engine-integrated SR (SRPO, RDG, FuseSR, MNSS, Deep Fourier), and recurrent VSR (RLSP, MRVSR, SSL), explaining why their engine coupling, fixed 4X scale, and runtime make them complementary rather than direct baselines for our 2X/3X, 1080p, post-decoder setting.

5. **Metrics, perceptual quality, and ablations (Secs. 4.1, 4.2, App. A.6).**
    We clarify how PSNR/SSIM/VMAF/LPIPS are computed, connect them to our 15-participant user study, and highlight existing ablations of ConvLSTM, PixelUnshuffle, and reparameterization, as well as our updated discussion of GPU utilization and frame-wise operation.

More details about the revisions are provided below:

* **Positioning of GameSR:** We clarify the positioning of GameSR in the revision. Our goal is to support **both local rendering and cloud gaming**, but always in an **engine-agnostic** way: GameSR runs purely on RGB frames and does not require access to G-buffers, motion vectors, depth, or vendor-specific hardware features (e.g., DLSS on RTX with dedicated integration). This makes it applicable to a much broader set of titles than current industrial upscalers: it can be used with legacy games such as TF2 on the original Source engine, with engines that do not expose the rendering buffers or motion/depth signals that modern DLSS/FSR plugins assume, and with cloud-gaming pipelines where only decoded video is available and transmitting extra render data would be too costly. Our **main comparison focus** is therefore on widely deployed game upsamplers (DLSS/FSR) because they represent the dominant practical baselines for high-quality gaming SR today; GameSR is designed to *complement* these methods by offering a vendor- and engine-independent alternative that works on both local PCs/consoles and cloud clients, including for titles and platforms where engine-integrated solutions are not feasible.


* **On the practicality of GameSR and the advantage of an engine-free design:** We agree that engine-level signals (G-buffers, motion, depth) are very powerful and that for AAA titles with existing DLSS/FSR-style integrations the quality–integration trade-off is often acceptable. GameSR, however, is explicitly targeted at cloud gaming and legacy or multi-engine catalogues, where clients only see encoded/decoded RGB frames and engine buffers are either inaccessible (closed-source, older engines) or too costly to transmit at scale. In these settings, just integrating it into the engine is not realistic, so an engine-agnostic post-decode upscaler is required.

---

> ### Author Response · Authors · 2025-12-02
>
> * **Comparisons against other works mentioned by the reviewers:**
>
>   * Our setting is fundamentally different from **GameStreamSR**, so we do not view it as a directly comparable baseline. GameStreamSR is an engine-dependent, depth-guided RoI upscaler: it applies a heavy SR network (instantiated with existing models such as EDSR) only to a small depth-defined region, and upscales the rest of the frame with bilinear interpolation. This design is tailored to mobile cloud gaming with access to render buffers and control over the engine, but leaves large portions of the image at bilinear quality (including HUD, distant targets, and text) and cannot be used in generic cloud-gaming clients that only see encoded/decoded RGB frames. In contrast, GameSR is a lightweight, full-frame post-render, post-decoder upscaler that operates purely on RGB video and can be deployed on both local and cloud clients, including legacy and closed-source titles where engine data are unavailable.
>
>   * **edge-SR** is built around very shallow (often 1–3 layer) single-image models, designed as drop-in replacements for classic upscalers on generic natural images. Modern 3D game frames are far more complex (high motion, fine geometry, HUDs, compression artifacts), and even **deeper** state-of-the-art SR baselines in our experiments struggle to fully recover accurate details. Given this, we do not expect a 1–3 layer image-only architecture to be competitive in our high-resolution, temporally demanding games.
>
>   * **SRPO** (ECCV’22), **RDG** (CVPR’25), **FuseSR**, **MNSS**, and **the Deep Fourier-based arbitrary-scale SR** method all target a different operating point than GameSR: they are tightly integrated into the real-time renderer and rely on internal engine signals such as high-resolution G-buffers, motion vectors, depth, and other per-pixel attributes. This requires modifying or instrumenting the game engine and is mainly practical when one controls the renderer (e.g., in-house AAA titles). By contrast, GameSR is deliberately designed as a **post-render, post-decoder** upscaler that operates solely on RGB frames: it can run (i) after video decode in a cloud-gaming client, and (ii) after a lower-resolution render target is resolved locally (e.g., 540p/720p→1080p) without any access to engine buffers or vendor-specific hardware. This makes it applicable to legacy and closed-source games, engines without forward-rendering data exposed, and generic streaming platforms where only decoded video is available. Moreover, these engine-coupled methods are typically evaluated in controlled rendering demos or author-created scenes rather than on long, recorded gameplay streams under encoding and network constraints as in our experiments. For these reasons, we view them as complementary engine-integrated approaches and do not treat them as directly comparable baselines for the post-render/post-decoder deployment scenario that GameSR targets.
>
>   * Regarding *Efficient Video Super-Resolution through Recurrent Latent Space Propagation* (**RLSP**), *Stable Long-Term Recurrent Video Super-Resolution* (**MRVSR**), and ***Structured Sparsity Learning for Efficient Video Super-Resolution***, we agree that they are important recurrent VSR baselines, but they target a very different domain than ours. All three are fixed x4 models evaluated on small LR inputs (typically 180×320→720p), whereas in gaming and modern upscalers such as DLSS/FSR the practical setting for 1080p is closer to 2x (e.g., 960×540→1080p or 720p→1080p); pushing to 4x from very low LR (e.g., 270p→1080p) is generally avoided because quality and object visibility degrade noticeably in games. A single fixed 4x scale also offers limited flexibility for any intelligent rate controller that needs to adapt rendering resolution to network congestion (e.g., switching between 360p/540p/720p with corresponding 2x/3x upscaling factors), which a 4x-only design cannot support. In addition, the reported runtimes for these models are on the order of tens of milliseconds per frame at 180×320 on high-end GPUs (TITAN RTX, TITAN Xp, V100), whereas GameSR achieves 2x/3x upscaling for full-HD gaming streams (540p/720p→1080p) in about 4–5 ms per frame on a similar GPU in terms of power with only 138K parameters. Adapting these architectures to our 1080p, 2x/3x, 60 fps setting would require substantial redesign and still provide less control to an adaptive rate controller, so we treat them as complementary general VSR work rather than directly comparable baselines for the deployment we target.

---

> > ### Author Response · Authors · 2025-12-02
> >
> > * **Expanded Evaluation:**
> >
> >   * **E2E Cloud Gaming System:** Our original submission focused on the SR component in isolation, but we have, in fact, implemented and evaluated a complete end-to-end cloud gaming system built on aiortc/WebRTC. The system includes an RL-based Joint Optimizer with a Complexity Analyzer on the server that selects the rendering resolution and encoder rate once per group of pictures (GOP), and GameSR running on the client-side after H.264 decode. In the revised paper, we add Sec. 4.3, “End-to-End Cloud Gaming System”, where we summarize this design and report results over 50 gaming sessions of CS2 and OW2 under realistic latency traces and two bandwidth regimes (30 Mbps and 8.5 Mbps). Compared to baseline WebRTC, our framework achieves savings of up to 50%, 62%, and 41% in bandwidth, CPU, and GPU requirements, respectively, while maintaining VMAF close to 90 and significantly improving quality at 8.5 Mbps. A separate subjective user study with 15 participants and approximately 200 played sessions further shows large MOS gains of up to 38% at 8.5 Mbps, and at 30 Mbps, our system maintains similar perceived quality while reducing bandwidth by about 50%, with MOS improving by 3.4% (OW2) and 4.6% (CS2) relative to baseline WebRTC. All these measurements include rendering, capture, encoding, network transmission, decoding, and GameSR upscaling; this indicates that the complete pipeline with GameSR integrated can satisfy real-time cloud gaming constraints in practice. Additional details are provided in Appendix A8.
> >
> >   * **Dataset Generalization & Scalability:** We appreciate the concern about game and genre diversity and agree that this is an important dimension. In this work, we chose to focus on three shooter titles and to train GameSR per game, similar to how DLSS and FSR are typically tuned per title in practice. We also understand that CS2, OW2, and TF2 may feel similar, since they are all shooters. However, from a rendering and content perspective they are quite different: CS2 is a realistic tactical FPS on the Source 2 engine with high-contrast lighting and fine geometric detail; Overwatch 2 is a fast-paced hero shooter on Blizzard’s Overwatch engine with highly stylized characters and dense particle effects; and TF2 is an older class-based shooter on the original Source engine with flatter shading and heavy HUD elements. In the revision, we also add a fourth title, FIFA, a third-person sports game with long-range camera views, large uniform textures (pitch/grass), and different motion patterns (passes, shots, camera pans), and report its results in Sec .\~4.2 (Table\~2). As shown there, FIFA follows the same trend as our other games in terms of quality and efficiency, illustrating that GameSR extends naturally beyond FPS titles and that adding a new game requires only training on that game’s footage. In addition, these four titles span engines with very different support for industrial upscalers: recent engines where DLSS/FSR-style plugins are available, engines that only support FSR-like spatial upscalers because the forward-rendering data required by DLSS-style methods is not exposed, and a legacy engine (TF2) where neither DLSS nor modern FSR variants can be deployed. One might argue that heuristic upsamplers such as bicubic, bilinear, or Lanczos are sufficient for such older games, but our results in Table\~2 show that even for TF2, the bicubic baseline is consistently weaker than GameSR across all scales, both in reconstruction quality and perceptual metrics.
> >
> >   * **Inclusion of 2K & 4K results:** We focus on 1080p because, according to the Steam Hardware & Software Survey (October 2025), it is still the dominant gaming resolution (≈53% of players), while 1440p and 4K account for ≈20% and ≈5%, respectively, so 1080p remains the most practical operating point for real-time cloud gaming and streaming. To directly address the concern about higher resolutions, we have added 2K and 4K experiments in Sec. 4.2 (“Scalability and Generality to High-Resolution Upscaling,” Table 3): GameSR reaches up to 39.25 dB PSNR, SSIM \> 0.999, and VMAF \> 93 for 1080p→4K while keeping inference \< 16 ms per frame (real-time 60 FPS), and only 7 ms per frame for 720p→2K (≈143 FPS, suitable for 120 Hz). These results are obtained on unseen 2K/4K clips collected after training, showing that a model trained solely at 1080p **generalizes and scales** to substantially higher resolutions without fine-tuning, while maintaining high perceptual quality and low latency for practical high-resolution cloud gaming scenarios.

---

> > > ### Author Response · Authors · 2025-12-02
> > >
> > > * **Evaluation Metric:** We apologize for the lack of clarity and will make the metric pipeline explicit. For all objective metrics (PSNR, SSIM, VMAF, LPIPS), the **high-resolution reference** is always the native game capture at full resolution (1080p in the main experiments, and 2K/4K in Sec. 4.2). The **low-resolution inputs** to GameSR are generated by downscaling these native captures: for 2x experiments we use 540p→1080p, for 3x we use 360p→1080p (and similarly 720p→2K, 1080p→4K, 720p→4K in the scalability experiments); in the cloud-gaming RD experiments, these LR sequences are additionally encoded and decoded to mimic a real streaming pipeline before being fed to GameSR. We then compute frame-wise PSNR/SSIM between the GameSR output and the native HR reference and report averages over each clip and game. VMAF is computed using the official Netflix implementation on the HR reference and the upscaled sequence, and LPIPS is computed using the standard PyTorch implementation; both have been widely shown to correlate well with human perception. Finally, our user study with 15 participants (Table 4\) yields high MOS scores (≈4.7/5) and consistent ratings across gamer/non-gamer groups, which supports that our VMAF/LPIPS trends indeed reflect perceived visual quality.
> > >
> > >   * **Hardware Scalability:** We apologize for the confusion and will clarify the intended hardware targets in the revision. GameSR is designed as a client-side upscaler for commodity cloud-gaming devices ranging from smartphones to desktop GPUs, not only desktops. To make this explicit, we have added Sec. 4.2 (“Inference Evaluation Across Heterogeneous Hardware,” Fig. 6(c), where we benchmark both the original GameSR and a mobile-oriented variant, GameSR-M, across devices from an Adreno 750–class Samsung S24 mobile GPU (4.7 TFLOPS) up to RTX 40-series desktop GPUs. On the Adreno 750, GameSR-M already achieves real-time performance with 26.56 ms (X2) and 14.55 ms (X3) per 1080p frame, demonstrating feasibility on modern handhelds where NPUs/GPUs are typically underutilized during cloud gaming. On discrete GPUs, latency scales roughly inversely with compute: mid-range cards (RTX 4060 Ti / 4070 Ti / A4000) sustain 1.7–4.0 ms per frame, while high-end RTX 3090/4090 reach 2.3 ms and 1.3 ms, respectively. These results show that our design is explicitly hardware-scalable and can even run on resource-constrained devices.
> > >
> > > * **Clarifications:**
> > >   * **GPU utilization:** In the configuration used for Fig. 4(c), the measured frame rates were approximately 123 FPS (native 1080p), 140 FPS (DLSS), 145 FPS (FSR), and about 125 FPS for GameSR. Thus, GameSR achieves essentially the same FPS as native while using only ≈82% GPU utilization, whereas native/DLSS/FSR run at similar or higher FPS but keep the GPU near 99–100% utilization. Both DLSS and FSR internally render at reduced resolution and then upscale to display resolution, and in typical usage the recovered shading budget is spent on higher FPS rather than reducing GPU load; this is consistent with our original text that “*DLSS/FSR are intended to boost framerate rather than reduce computation.*” Our point in Fig. 4(c) is therefore not that DLSS/FSR are inefficient or that high GPU utilization is inherently negative, but that the GameSR configuration can match native frame rate while leaving more GPU headroom. In a cloud-gaming deployment, where rendering is done on the server and upscaling is done on the client, this headroom can be converted into lower server-side rendering cost and reduced streaming bitrate by rendering at lower resolution and adding only a small SR overhead on the client. We have updated the main text around Fig. 4(c) to explicitly report these FPS numbers together with the utilization trends.
> > >
> > >   * **Ablation study:** We already provide an ablation in Sec. 4.2 and App. A.6 (Table 7), where we remove ConvLSTM, PixelUnshuffle, and reparameterization and jointly measure PSNR/SSIM/LPIPS, latency, parameters, and memory. For example, removing ConvLSTM reduces latency to 3.05 ms and parameters to 65K but degrades PSNR by about 5 dB (37.99 → 32.99), and the ConvLSTM hidden state is kept locally on the client and never transmitted, so it adds no bandwidth overhead. We will surface these key numbers more prominently in the main paper.
> > >
> > >   * **Key/delta frames:** In our cloud-gaming setting, the client receives a live stream of decoded RGB frames, so GameSR is designed to operate per frame and is agnostic to codec internals such as I/P/B frame types. Given its low latency (approximately 3–6 ms per 1080p frame on mid-range GPUs), there is no need to upscale only keyframes and interpolate the rest, a strategy that is more suitable for offline VOD pipelines than for interactive, high-motion gaming.

---

### Meta-Review · Area_Chair_UjUZ · 2025-12-12

**Summary:**

Unanimity of the reviewers on the negative side. The authors provided responses to a number of concerns.
The ACs carefully read the paper, the reviews, and the authors' responses.
The paper requires major revision and a second review round.

**Reviewer Concerns:**

Unanimity of the reviewers on the negative side. The authors provided responses to a number of concerns.
The ACs carefully read the paper, the reviews, and the authors' responses.
The paper requires major revision and a second review round.

**Reviewer Scores:**

Unanimity of the reviewers on the negative side. The authors provided responses to a number of concerns.
The ACs carefully read the paper, the reviews, and the authors' responses.
The paper requires major revision and a second review round.

---

### Decision · Program_Chairs · 2026-01-26

Reject